# Stimulus dependence of directed information exchange between cortical layers in macaque V1

**Marc Alwin Gieselmann, Alexander Thiele***

Biosciences Institute, Newcastle University, Newcastle upon Tyne, United Kingdom

**Abstract** Perception and cognition require the integration of feedforward sensory information with feedback signals. Using different sized stimuli, we isolate spectral signatures of feedforward and feedback signals, and their effect on communication between layers in primary visual cortex of male macaque monkeys. Small stimuli elicited gamma frequency oscillations predominantly in the superficial layers. These Granger-causally originated in upper layer 4 and lower supragranular layers. Unexpectedly, large stimuli generated strong narrow band gamma oscillatory activity across cortical layers. They Granger-causally arose in layer 5, were conveyed through layer six to superficial layers, and violated existing models of feedback spectral signatures. Equally surprising, with large stimuli, alpha band oscillatory activity arose predominantly in granular and supragranular layers and communicated in a feedforward direction. Thus, oscillations in specific frequency bands are dynamically modulated to serve feedback and feedforward communication and are not restricted to specific cortical layers in V1.

## Editor's evaluation

The cortical column can be considered a mesoscopic scale unit of computation. While neural anatomy suggests a well-known flow of information from granular to supra-granular to infra-granular layers, how well this maps onto feedforward and feedback signal flow in different frequency bands is less clear. In this paper Gieselmann and Thiele show that stimulus properties play a crucial role in determining the flow.

**\*For correspondence:**
alex.thiele@newcastle.ac.uk

**Competing interest:** The authors declare that no competing interests exist.

## Introduction

The multiple layers of the neocortex show an elaborate connectivity pattern, which predicts a specific directionality of interlaminar communication (*Callaway, 1998*; *Douglas and Martin, 2004*; *Douglas et al., 1989*). Feedforward inputs predominantly terminate in layer 4 of sensory cortical areas, before being passed to layers 2/3 and then onwards to layers 5/6, where recurrent inputs to layer 2/3 arise (*Callaway, 1998*; *Callaway, 2004*). Conversely, feedback connections from downstream areas predominantly terminate in layer 1 and 5 (*Rockland and Pandya, 1979*), from where the information is passed to layers 2/3. Local field potential (LFP) recordings, using different stimulus and task conditions, have provided physiological support for these anatomical pathways. Current source density analysis of the LFP, as well as latency of spiking activity has demonstrated that the earliest activity elicited by stimuli centered on the classical receptive field occurs in layer 4c (*Bijanzadeh et al., 2018*). Stimuli restricted to the far receptive field surround, which are hypothesized to affect processing in V1 primarily through feedback projections (*Bair et al., 2003*), initially activate V1 in upper superficial layers and in layer 5 (*Bijanzadeh et al., 2018*). In addition to this anatomical segregation of feedforward and feedback signal input to V1, it has been proposed that feedforward and feedback signals have separable

spectral signatures of the LFP. Feedback processing has been aligned with low frequency (alpha [α] band) oscillatory activity, dominating activity in infragranular layers (*Buffalo et al., 2011*; *Lopes Da Silva and Storm Van Leeuwen, 1977*; *van Kerkoerle et al., 2014*), and with beta (β) band oscillations (*Bastos et al., 2015a*; *Xing et al., 2012*). Conversely, feedforward processing has been associated with low frequency theta [θ], and higher frequency oscillatory activity (gamma [γ] band), originating in layer 4, dominating in (or even being largely restricted to) supragranular layers (*Bastos et al., 2015a*; *Bollimunta et al., 2011*; *Buffalo et al., 2004*; *Buffalo et al., 2011*; *Maier et al., 2010*; *Spaak et al., 2012*; *van Kerkoerle et al., 2014*; *Xing et al., 2012*). However, the generality of this scenario has recently been questioned (*Haegens et al., 2015*), whereby the strongest low-frequency (α) oscillations generators were found in supragranular layers of sensory cortex. Moreover, while most studies failed to find prominent γ oscillations outside supragranular layers (*Buffalo et al., 2011*; *Maier et al., 2010*; *Xing et al., 2012*), exceptions do exist (*Maier et al., 2011*).

To address these controversies and investigate the origin and spectral signature of feedforward and feedback signals in detail, we recorded across cortical layers of V1 of awake macaque monkeys, while manipulating stimulus size systematically. Increasing the stimulus size invokes center surround interactions, which result in specific input patterns across a cortical column (*Bijanzadeh et al., 2018*), and which strongly alter the spectral signatures LFPs. Increasing the size of a patch of an orientated grating centered on a receptive field (RF) increases γ power in conjunction with a general decrease of neurons firing rate (*Bauer et al., 1995*; *Gieselmann and Thiele, 2008*; *Jia et al., 2011*; *Ray and Maunsell, 2010*). If this increased γ power was a signature of a feedforward signal, then we would expect it to arise in either layer 4, or layer 2/3 of V1. Conversely, the feedback signals should show spectral signatures in lower frequency bands, such as α (*van Kerkoerle et al., 2014*), which should arise and dominate in layers 1 and 5 (*Bastos et al., 2015a*; *van Kerkoerle et al., 2014*).

While our results support some of the features outlined above, they violate them in multiple aspects, and show that feedforward and feedback signals attain stimulus dependent oscillatory frequencies across cortical layers on V1.

## Results

We recorded neuronal activity from area V1 from 16 contact laminar electrodes in two passively viewing male macaque monkeys. Electrodes were inserted perpendicular to the cortical surface, aiming to record from within a single V1 cortical column. To ensure consistency of our analysis we rejected experiments (total n = 25/9, monkey 1/2) from further analysis if the current-source-density (CSD, see Materials and methods) profiles did not meet any of the following three conditions: (1) An identifiable current source as early as 50ms after the stimulus onset (n = 1/1, monkey 1/2). (2) The 'early' current source is identified on a channel between the seventh and twelfth contact (from the top of the electrode array, to ensure a minimum coverage of cortical layers 2–6; n = 3/1, monkey 1/2). (3) A pattern of the CSD-profile that is consistent with the majority of experiments (n = 1/1, monkey 4/0). After applying the rejection criteria, we were left with 24 experiments (monkey 1: n = 17, monkey 2: n = 7) from three hemispheres which were included into this study.

*Figure 1* shows the alignment of neuronal data from an example experiment. The data were obtained during receptive field (RF) mapping (*Figure 1B*) and during presentation of gratings (*Figure 1C and D*). In this recording the eighth channel from the top was defined as the alignment channel indicating putative layer 4c at zero depth (*Figure 1A*). Correspondingly, the RF maps of the envelope of multi-unit-activity (MUAe, see Materials and methods) showed the early emergence of a visual RF in the same channel between 30 and 50ms (*Figure 1B* left), demonstrating that the early current sink coincides with the earliest activation from thalamo-cortical afferents. This electrode placement resulted in contacts in supragranular layers (SG), alignment layer 4c, and infragranular layers (IG) (*Figure 1A*). The gray matter region (from third channel from top to sixths from the bottom in *Figure 1A*) is flanked by two large current sources at the top and at the bottom (*Figure 1D*). The perpendicularity of probe orientation relative to the cortical surface is indicated by the good spatial alignment of RF-centers (*Figure 1B*). The LFP responses in the gray matter region show strong evoked potentials, and sustained oscillations (*Figure 1C*). The lower five channels in *Figure 1* were located in the white matter, and possibly in area V2, as evident by a sudden shift in visual RF's (*Figure 1B*, lowest contacts). While the strength of the LFP-signal seems to be unchanged (*Figure 1C*) for these contacts,

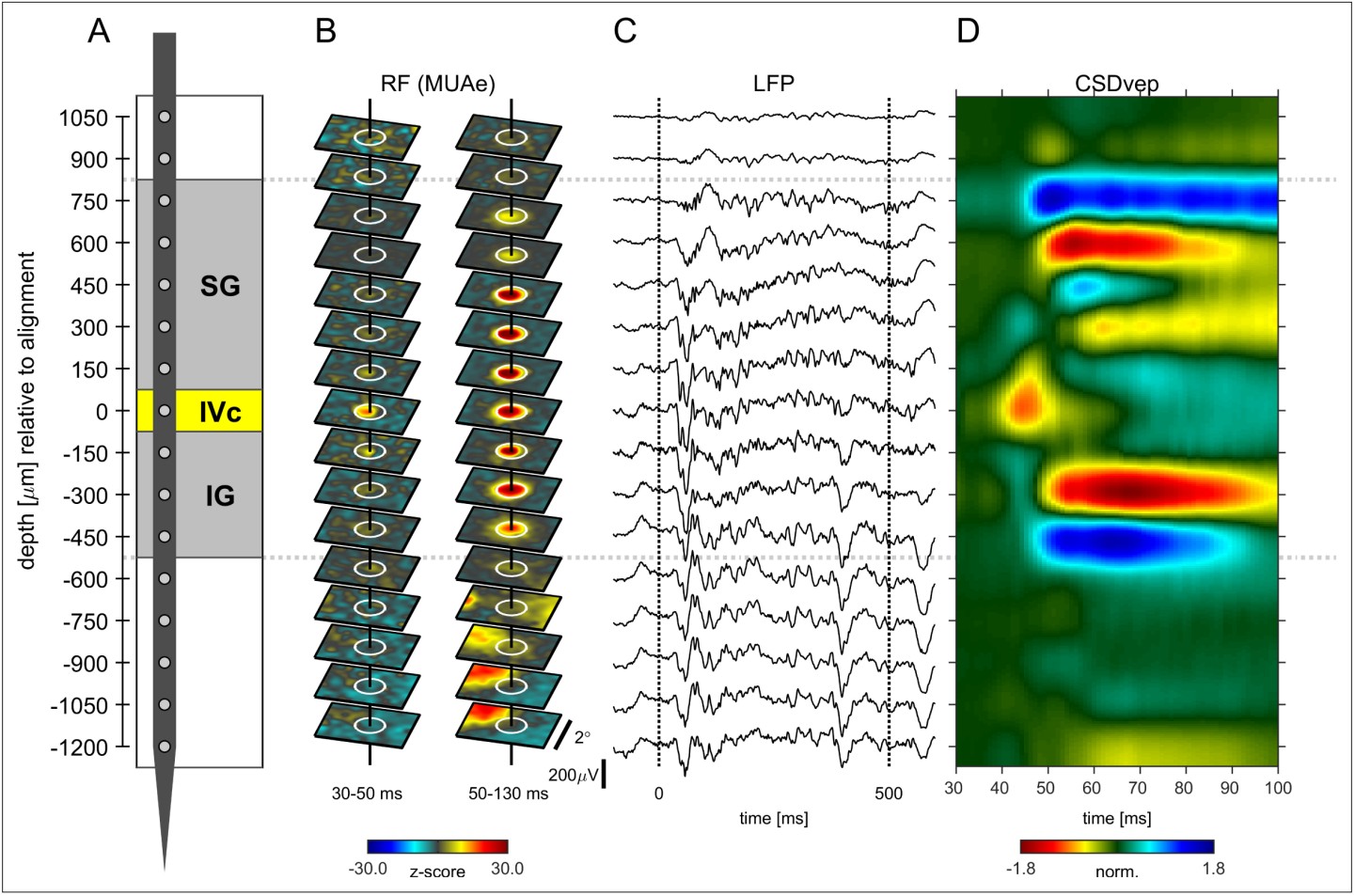

**Figure 1.** Example experiment (**A**) Illustration of probe position and electrode contacts in relation to cortical gray matter during a recording session. Position in depth is aligned to input layer 4c at 0 μm. Gray dotted lines show boundaries of neuronal responsivity across panels (**A–D**). (**B**) Minimum response fields of MUAe for each channel. Vertical solid black line shows RF center, white circle indicates a 1° diameter. Left column shows low latency response between 30 and 50ms after stimulus onset. Right column shows response between 50 and 130ms after stimulus onset. Activity is z-scored to baseline period. (**C**) Response profile of LFP activity for a single trial with a stimulus of 3° in diameter. (**D**) Depth profile of the CSD signal averaged across all stimulus presentations. Red colors indicate current-sinks and blue colors indicate current-sources. Colormap represents CSD amplitude normalized to the positive peak at 0 μm between 40 and 60ms.

The online version of this article includes the following figure supplement(s) for figure 1:

**Figure supplement 1.** Analysis of minimum response fields derived from RF-mapping paradigm.

current sources and sinks were weaker and less structured than in the gray matter. A detailed analysis of RF sizes for different layers is presented in *Figure 1—figure supplement 1*.

We next focus our analysis on basic phenomena of spiking activity and current source density changes with different sized stimuli, as these provide insight into the underlying pathways that mediate changes (feedforward, lateral intra-areal, intracolumnar feedback and feedback from downstream areas), as well as providing a benchmark against previously published results.

## Response latency as a function of cortical depth and stimulus size

We calculated spiking latencies by using the MUAe signal using a curve-fitting procedure (*Figure 2A*, see Materials and methods and *Self et al., 2013*). The shortest latencies occurred in granular layers, followed by infragranular layers, with supragranular layers generally lagging behind (*Figure 2B*). This overall pattern was independent of stimulus size used. However, stimulus size significantly altered response latencies in granular ($p < 0.01$, one-factor-ANOVA, *Figure 2C*) and in infragranular layers ($p < 0.001$, one-factor-ANOVA, *Figure 2C*). In granular layers larger stimuli resulted in up to ~3ms longer latencies, while in infragranular layers they caused up to ~6ms shorter latencies. Thus, when

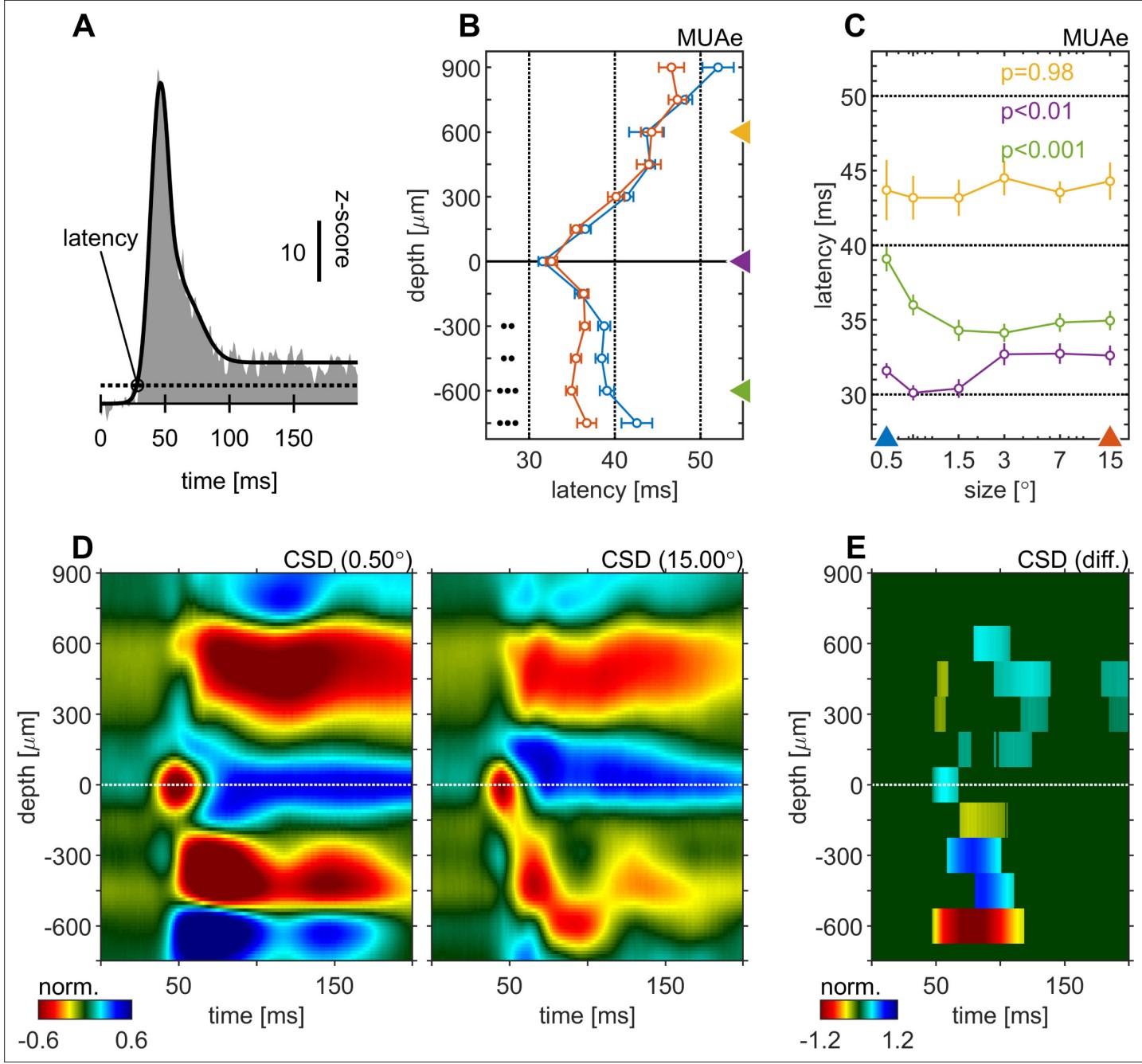

**Figure 2.** Centre surround effects on early responses of MUAe and CSD depth-profiles. (**A**) Initial MUAe response used to derive response latencies across depths. The example is from the alignment channel of the experiment shown in *Figure 1*. (**B**) Depth-profiles of MUAe latency. Black dots indicate significant levels of latency differences (signed-rank test, •••: p < 0.0001, ••: p < 0.001, •: p < 0.01; FDR-corrected) between large (red, 15°) and small (blue, 0.5°) stimulus. (**C**) Size-tuning of MUAe latency for three laminar compartments (yellow: supragranular, purple: granular, green: infragranular). Latencies from supragranular and in infragranular layers were averaged across two adjacent channels (see triangles on depth in B). p-Values indicate size dependent differences in latency for the three contacts (ANOVA). (**D**) CSD depth-profile for a small (left, 0.5°) and a large (right, 15°) stimulus, averaged across experiments. Profile of each session was normalized to the amplitude of the early current sink in putative layer 4c (pos. peak at 0 µm between 40 and 70ms). (**E**) Depth-profile showing significant differences (large minus small) between the two depth-profiles in D (p < 0.05, cluster-mass-test).

The online version of this article includes the following figure supplement(s) for figure 2:

**Figure supplement 1.** Centre surround effects on early responses of MUAe and CSD depth-profiles.

**Figure supplement 2.** Centre surround effects on early responses of MUAe and CSD depth-profiles.

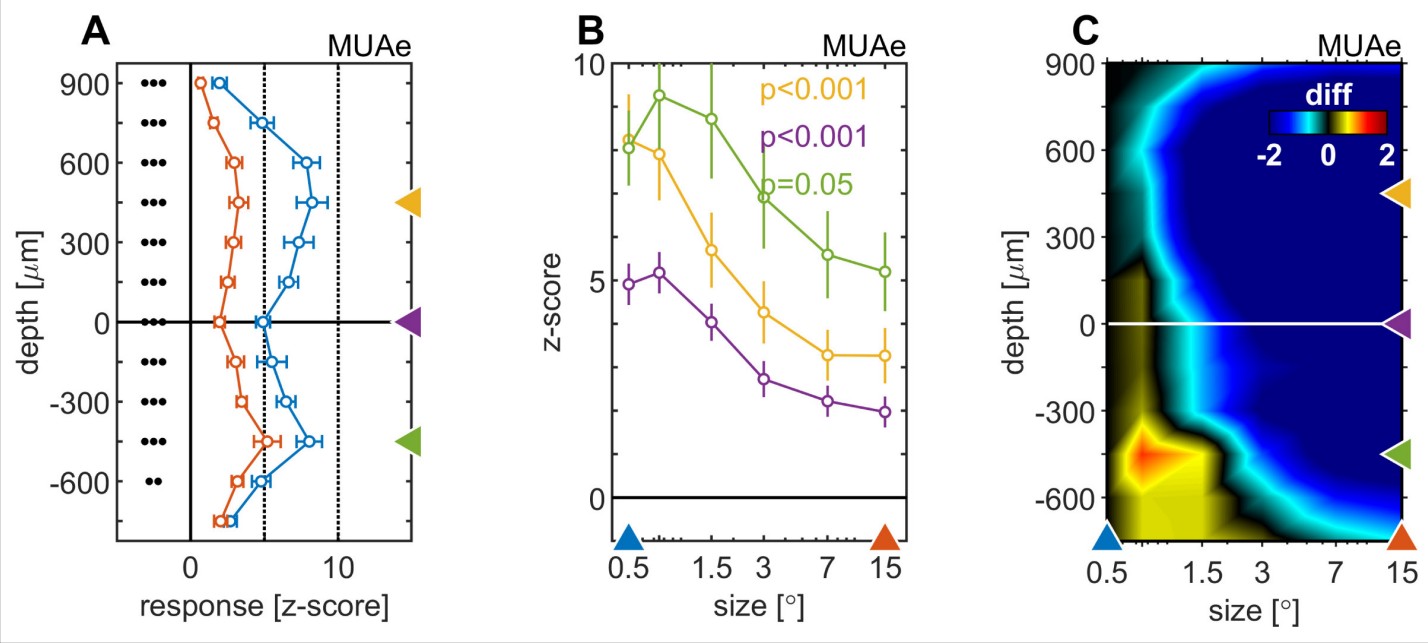

**Figure 3.** Centre surround effects on late/sustained (200–500ms) response properties of MUAe. (**A**) Depth-profiles show average z-scores (normalized to spontaneous activity) across experiments and error bars represent the s.e.m. for two selected stimulus sizes. Grouped black dots indicate significant levels of signed-rank test (●●●: p < 0.0001, ●●: p < 0.001, ●: p < 0.01; FDR-corrected) between small (0.5°, blue) and large (15°, red) stimulus diameter (**B**) Size-tuning and response-depth-profiles of MUAe data for all grating diameters and three laminar locations. Yellow, green, and purple lines/symbols refer to different depths in the supragranular ( + 450 μm), infragranular (–450 μm) and granular (alignment, 0 μm) regions. (**C**) Surface plot of the differences of MUAe z-score response relative to reference size 0.5°.

large stimuli were presented the lag between granular and infragranular layers was reduced to ~2ms (from ~7ms for small stimuli, *Figure 2C*). No significant changes were found for supragranular layers (p = 0.98, one-factor-ANOVA, *Figure 2C*). Latency changes also appeared to occur in the top-most layer, although this was not significant, possibly because of relatively small sample sizes (not every recording included that layer/contact due to electrode placement).

The CSD profile equally changed with stimulus size. *Figure 2D and E* show the average CSD profile across all experiments for a small (0.5°) and a large (15.0°) stimulus. The CSD-profile of each experiment was normalized to the peak of the early onset current sink in the alignment layer (0 μm). *Figure 2E* shows a surface plot of time depth CSD differences plot (difference between CSD profiles in C and D), where only significant differences are displayed (p < 0.05, cluster-mass-test). In granular and supragranular layers we observed moderate decreases in inward current amplitudes with larger stimuli, which occurred as early as 50ms in granular layers. Most of the changes in and above layer 0 can be described as a general attenuation of current amplitudes. In other words, the overall laminar pattern of current sinks and sources in granular and supragranular layers was similar in the CSD profile for small and large stimuli, but it was overall slightly attenuated for large stimuli, probably due to surround suppression effects at the level of the lateral-geniculate-nucleus (LGN). The currents (sinks) in infragranular layers were also mostly reduced, but in addition changes in the pattern of sinks and sources occurred. Specifically, a current sink emerged ~75ms after stimulus onset at –600 μm (*Figure 2D and E*), where a source was present when small stimuli were presented. This indicates that stimulating the surround of a V1 RF results in additional input to infragranular layers. Given that RF sizes differed somewhat between layers (*Figure 1—figure supplement 1*) it could be argued that the analysis should be repeated for different reference sizes (other than e.g. 0.5°). Doing so yielded qualitatively identical results (*Figure 2—figure supplement 1*; *Figure 2—figure supplement 2*).

We next focused on surround suppression/enhancement effects on MUAe, which are most profound during later parts of the response. Here we analyzed the activity from 200 to 500ms after stimulus onset. Surround suppression affected all cortical layers. However, it was more pronounced in supragranular and granular than in infragranular layers (*Figure 3A and B*). Moreover, the stimulus size that resulted in maximal activity equally varied between layers, with slightly larger stimuli (0.75°,

1.5°) eliciting maximal activity in infragranular layers, while the smallest stimulus (0.5°) elicited maximal activity in supragranular layers (*Figure 3C*).

Together the results from *Figures 2 and 3* are compatible with recent reports suggesting that stimuli extending into the far surround of neurons invoke center surround interactions through feedback connections from higher cortical areas that terminate predominantly in layers 1 and 5 (*Bijanzadeh et al., 2018*; *Nassi et al., 2013*; *Nurminen et al., 2018*; *van Kerkoerle et al., 2014*).

## LFP spectral power and phase relationship analysis

We now turn to the question how feedback signals impose their effects on local cortical processing in area V1. It has been argued that feedback signals can be characterized in terms of low frequency (α, β) oscillatory activity, while feedforward signals can be characterized in terms of θ- and γ-frequency oscillations (*Bastos et al., 2015a*; *Bosman et al., 2012*; *Buschman and Miller, 2009*; *Fries, 2015*). A simple prediction would thus be that all stimulus sizes elicit γ frequency oscillations in layers 2/3, as all of the stimuli will invoke feedforward drive. Within this context, two scenarios are possible. Firstly, γ frequency oscillations in L2/3 are slightly reduced when larger stimuli are presented, due to the effects of surround suppression that are inherited from the LGN. Alternatively, their spectral power increases as stimulus size increases, because large stimuli invoke stronger inhibitory surround interactions, which through pyramidal-inhibitory interactions drive γ frequency oscillations (*Tiesinga and Sejnowski, 2009*). Additionally, infragranular layers should show limited γ frequency power (*Dougherty et al., 2017*; *Maier et al., 2011*; *Spaak et al., 2012*; *Xing et al., 2012*), but they should show increased low frequency (α/β) oscillations with increasing stimulus size as the level of feedback input increases (*Bastos et al., 2015a*; *Bosman et al., 2012*; *Buffalo et al., 2011*; *van Kerkoerle et al., 2014*).

To answer these questions, we transformed the raw LFP signal into the bipolar referenced LFP (LFPbp, see Materials and methods). This was necessary, as the raw LFP is not a sufficiently local signal to allow for meaningful layer specific analysis (*Trongnetrpunya et al., 2016*). The single trial LFPbp of the sustained part of the response (200–500ms after stimulus onset) was then subjected to standard spectral analysis (see Materials and methods). We focused on the sustained part of the neuronal response, as endogenous γ oscillations are most pronounced during this period (*Gieselmann and Thiele, 2008*). The results are shown in *Figure 4*. It shows a CSD profile (*Figure 4A*) and the layer dependent induced spectral power (iSP, z-score relative to spontaneous activity 300ms before stimulus onset) as a function of stimulus size for an example experiment (*Figure 4B*, for cross-cuts at selected depths see *Figure 4—figure supplement 3*). When small stimuli were presented, the supragranular layers showed relatively strong iSP across multiple γ frequencies (20–100 Hz). In addition, there was weaker, but nevertheless pronounced broad band iSP (20–100 Hz) in infragranular layers. As stimulus size increased the iSP across all layers became more focused with clear peaks emerging in the low γ frequency range (~40 Hz). This pattern became even more pronounced when analyzing iSP across different laminae averaged across all experiments (*Figure 4C–E*, *Figure 4—figure supplement 3*). Small stimuli (0.5°) elicit broad band power changes ranging from 30 to 100 Hz, with increasing iSP toward higher frequencies in supragranular layers (*Figure 4D*). Infragranular layers showed relatively small, but~ homogenous broad band β/γ iSP in the range from 20 to 80 Hz (*Figure 4D*). When large stimuli were shown, iSP in the supragranular layers concentrated in more narrow band regions of ~40 Hz. A smaller peak in that frequency range also occurred in granular layers. A surprisingly large γ frequency iSP increase occurred in infragranular layers 5 and 6, at ~40–50 Hz (compare to *Xing et al., 2012*). Also, the peak frequency of the evoked gamma oscillation at around 40 Hz showed the same inverse relationship to stimulus size (*Figure 4—figure supplement 3* and *Gieselmann and Thiele, 2008*), and is in line with other previous publications using grating stimuli (*Bauer et al., 1995*; *Gray et al., 1989*; *Jia et al., 2011*). Calculating power changes relative to different reference stimuli resulted in similar outcomes.

We quantified the effects of stimulus size on spectral power by calculating the mean iSP of the LFPbp-signal in discrete frequency bands. This was done for the θ (4–8 Hz, *Figure 5A–C*), α (8–13 Hz, *Figure 5D–F*), β (14–25 Hz, *Figure 5G–I*) and low γ (35–55 Hz, *Figure 5J–L*) range. *Figure 5A, D, G and J* show the average depth profiles of band power in response to a small (0.5°, blue) and a large (15°, red) stimulus. For all frequency bands, stimulus size modulated the band,

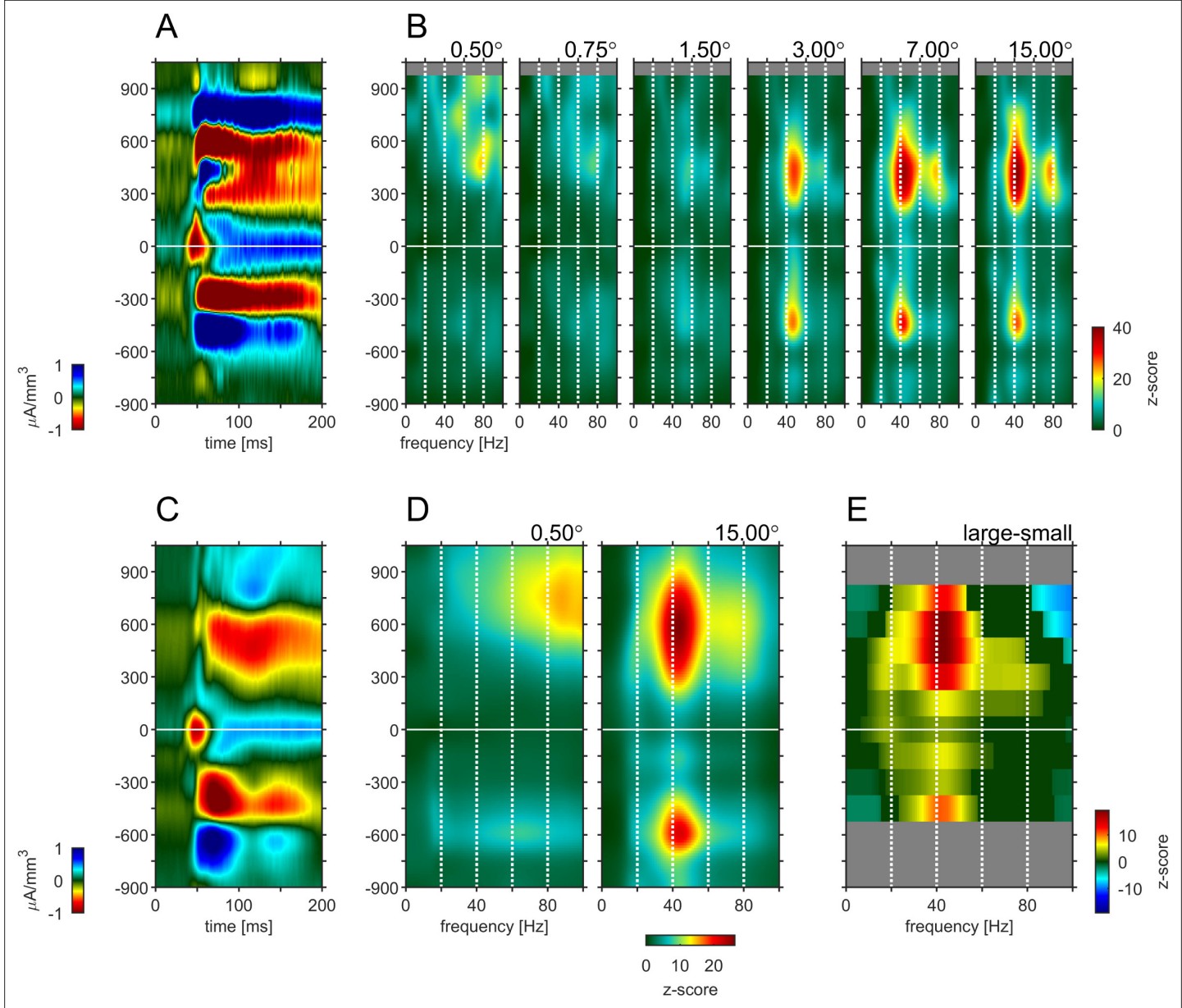

**Figure 4.** Depth-profile of spectral density for 'small' and 'large' stimuli. (**A**) Depth profile of the CSD signal for a 0.5° grating in the example experiment (see *Figure 1*). (**B**) Depth-profile of LFPbp-spectral-density (normalized to spontaneous activity as z-score) for all six grating diameter in the example experiment. (**C**) Depth profile of the CSD signal for a 0.5° grating across all experiments (n = 24). (**D**) Depth-profile of LFPbp-spectral-density for 'small' (0.5°) and 'large' (7.0°) gratings across all experiments. (**E**) Depth-profile of the difference in spectral-density between 'large' and 'small' gratings. Warm and cold colors indicate increases and decreases with stimulus size. Gray color indicates cortical depths which were not covered by all of the n = 24 experiments. Dark green color indicates non-significant differences according to a 'non-parametric cluster analysis' or 'cluster-mass-test' (*Bullmore et al., 1999*; *Maris and Oostenveld, 2007*; *Self et al., 2013*).

The online version of this article includes the following figure supplement(s) for figure 4:

**Figure supplement 1.** Depth-profile of spectral density for 'small' and 'large' stimuli.

**Figure supplement 2.** Depth-profile of spectral density for 'small' and 'large' stimuli.

**Figure supplement 3.** Average power spectrum at four representative depth positions relative to the alignment layer for each of the tested stimulus sizes.

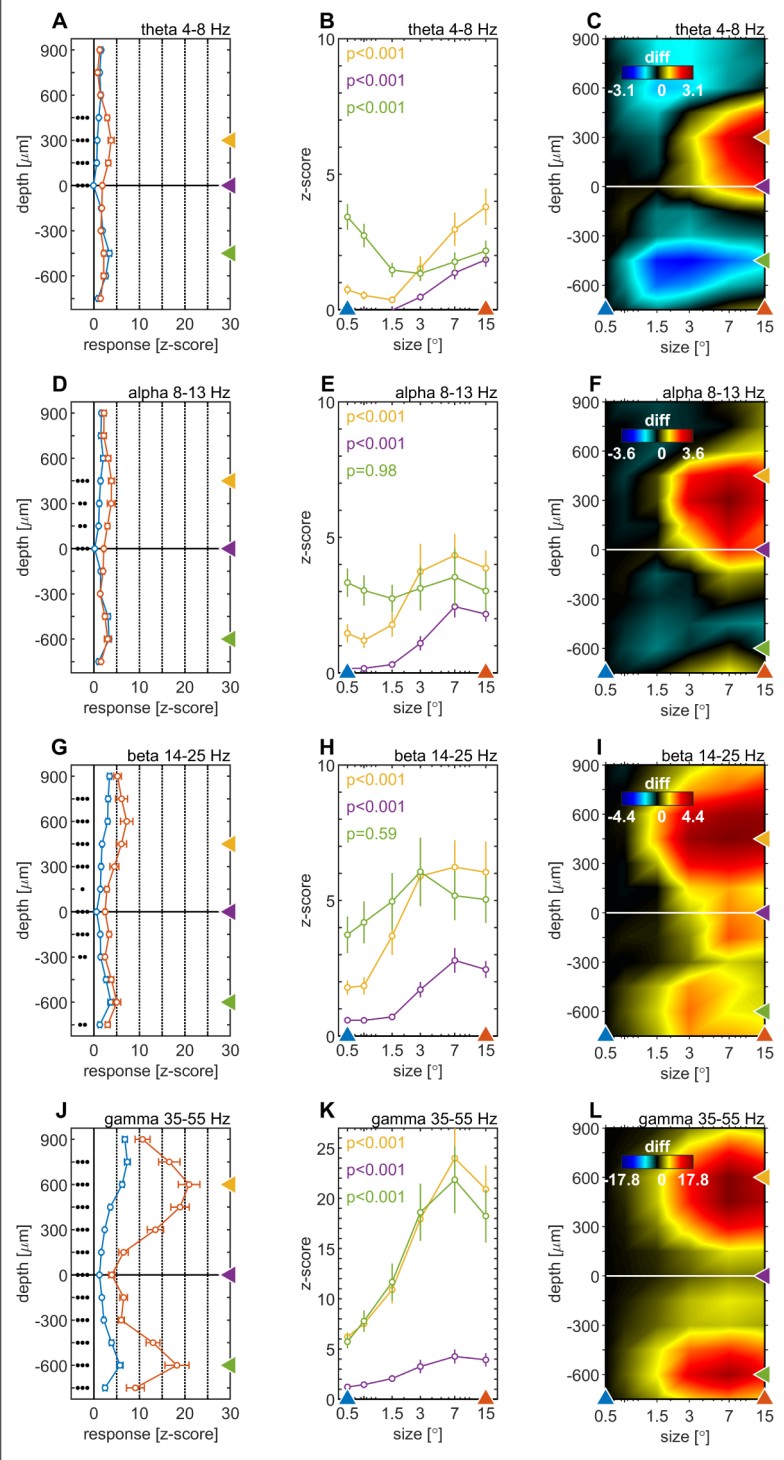

**Figure 5.** Centre surround effects on late/sustained (200–500ms) responses in 4 'classic' frequency bands. Panel rows show the analysis in four spectral bands, θ : 4–8 Hz (**A–C**), α: 8–13 Hz (**D–F**), β: 14–25 Hz (**G–I**), γ: 35–55 Hz (**J–L**). The first panel column (**A, D, G, J**) are response-depth-profiles of average power of LFPbp signal for small (0.5°, blue) and large (15°, red) stimuli. Grouped black dots indicate significant levels of signed-rank test (•••: p < 0.0001, ••: p < 0.001, •: p < 0.01; FDR-corrected). The second panel column (**B, E, H, K**) show size-tuning-curves of LFPbp spectra for all grating diameters and three laminar locations cherry-picked for each frequency band. Yellow, green and purple lines/symbols refer to depths in the supragranular, infragranular and granular (alignment, 0 μm) regions, respectively. The third panel column (**C, F, I, L**) shows color-map representations of the differences of LFPbp spectral response for each stimulus size relative to 0.5°.

but these modulations differed between layers and frequency bands. For the θ-band, α-band, and β-band, stimulus-size-induced changes are most pronounced for granular and lower supragranular layers (*Figure 5C, F* 1). For infragranular layers, larger stimuli caused a reduction in θ-band (*Figure 5B*), no change in α-band or β-band, and strong increases in γ-band. Overall, larger stimuli caused significantly more γ-band across all contacts (*Figure 5J*), but relative changes are strongest in supragranular and infragranular layers (*Figure 5L*).

## Gamma-connectivity across cortical laminae

Previous studies have argued that phase-locked oscillatory activity can provide insight into the directionality of information flow for a given frequency band. Specifically, it was shown that α-oscillations locked to the α-phase in granular layers appeared to originate in infragranular and upper supragranular layers, progressing toward middle supragranular layers (*van Kerkoerle et al., 2014*). Conversely, γ-oscillations locked to the γ -phase in granular layers originated in granular or lower supragranular layer progressing toward upper supragranular and infragranular layers respectively. To analyze this for different stimulus sizes, we computed the phase-triggered average (PTA) to estimate the phase-locked oscillations between electrode contacts for different frequency bands. Phase-triggered-averaging describes the alignment of LFPbp signals of a given electrode contact to the phase of an oscillation at a defined frequency at the reference-channel. The aligned LFPbp is then averaged across the number of alignments. Temporal structure in the resulting histogram can be interpreted as a phase-coupled oscillation between the two signals. *Figure 6* shows the depth profiles of the four main frequency band PTA-histograms for a reference channel at –150 μm, averaged across all experiments. As expected, the depth used as the reference phase of the oscillatory cycle always showed strong phase coupling, with close to zero phase delays (*Figure 6*, gray triangles indicate the reference channel). Many other contacts showed coupling to the reference channel for all frequency bands. However, the exact phase coupling varied between frequency bands and between stimulus sizes. Low frequency (θ, α) showed a phase progression from layer 5 to layer 6, and then to granular and supragranular layers for small stimuli. This pattern changed radically for larger stimuli, where phase progression was seen bidirectionally from granular to supragranular and to infragranular layer 5, but also from infragranular layer 6 to layer 5. For γ frequencies, layer 4 and 6 were largely in phase for small stimuli, but almost 90° out of phase with layer 5. Some continuous phase progression could be seen from granular toward upper supragranular layers. However, for large stimuli phase progression occurred from layer 5 to layer 6, onwards to granular layers, and then to supragranular layers. These patterns were not artefacts of selecting a specific reference channel (*Figure 6—figure supplements 1–4*). These patterns were less pronounced when using the iCSD signal instead of the bpLFP signal but the globally described pattern largely prevailed (*Figure 6—figure supplements 1–4*).

Isolated phase relationships per se do not yield insight into causal interactions, as phase lags can equally be interpreted as phase advances due to the periodicity of the signal. Moreover, causal interaction assignment based on phase relationships can be inconsistent with Granger causality analysis (*Brovelli et al., 2004*). To gain insight into causality, we performed spectral Granger-causality (GC) analysis (*Figure 7—figure supplement 1*). We then integrated the GC-Index in the four different frequency bands analyzed above (θ, α, β, and γ frequency) and generated connectivity plots using the net GC (difference between directions) for each connection (*Figure 7*). Hence, *Figure 7* shows the direction and associated strength of granger-causal influence of the four different frequency LFP bands between different layers. We only plotted those connections that passed the reverse-granger-test (RGT) to protect against spurious results induced by linear mixed noise (*Haufe et al., 2013*; *Vinck et al., 2015*). In *Figure 7—figure supplements 2 and 3* we show the same data including the connections which failed RGT and the results of the time reversed GC analysis, respectively. *Figure 7* shows the analysis of non-conditional GC. Conditional GC produced qualitatively equivalent results, albeit with overall smaller granger causality indices. A striking feature of the directionality of all GCs (irrespective of the frequency bands analyzed) was its dominance in the 'upward' direction, namely toward supragranular layers. This indicates that communication was directed toward the cortico-cortical feed-forward channels in V1 (for related results see *Ferro et al., 2021*). Sizeable GCs in the downward direction were usually restricted to granular to infragranular interactions, and to layer 5 to layer 6 interactions. Downward GCs were generally more pronounced with larger stimuli.

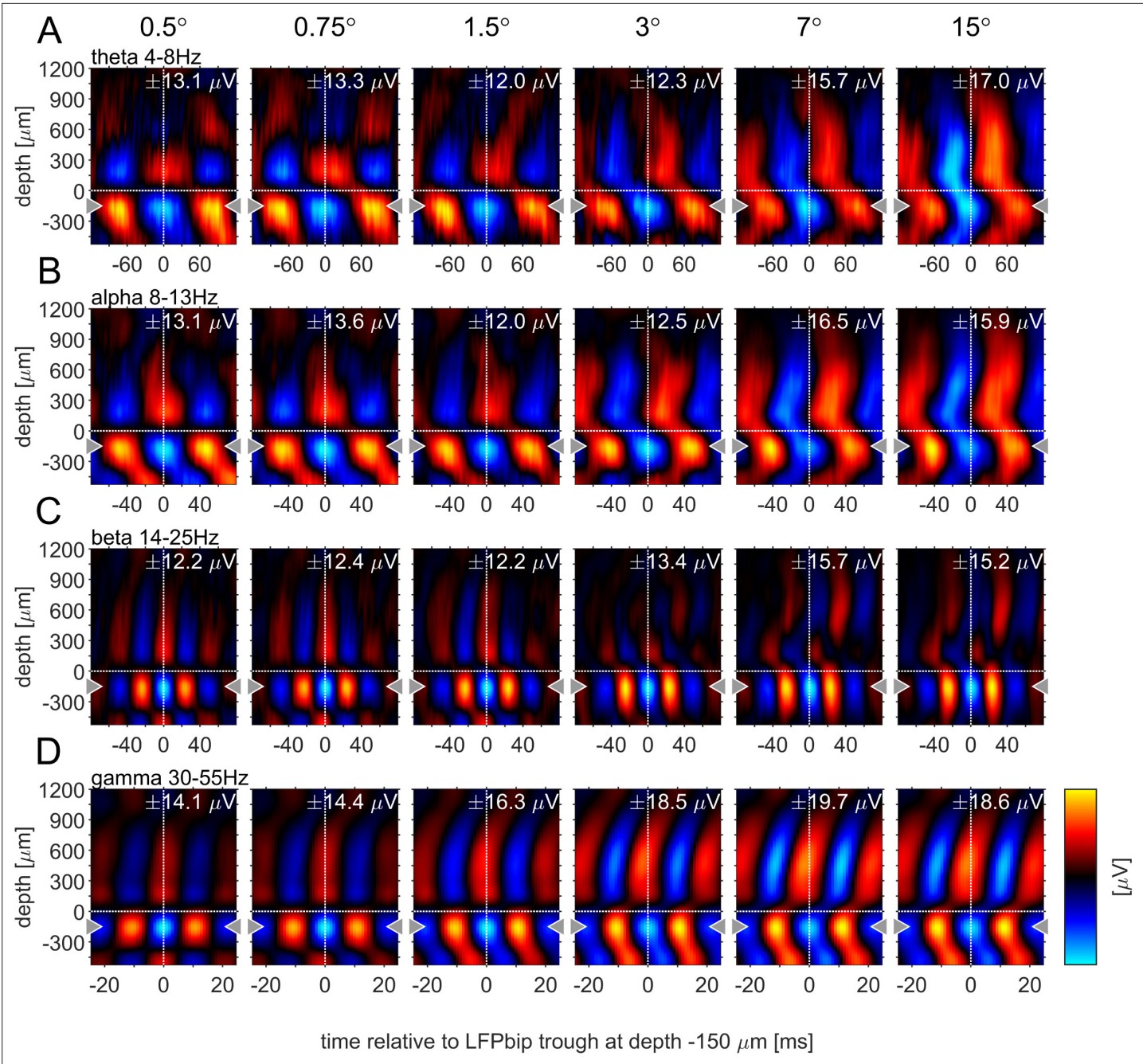

**Figure 6.** Phase-triggered-average (PTA) between cortical layers in four frequency bands. Laminar profiles of cortical oscillations of the LFPbp triggered by the phase of the LFPbp at the reference contact at −150 μm (indicated by the gray triangle). (**A**) Trigger LFP $\theta$-band filtered [4–8 Hz]. (**B**) Trigger LFP α-band filtered [8–13 Hz]. (**C**) Trigger LFP β-band filtered [14–25 Hz]. (**D**) Trigger LFP γ-band filtered [35–55 Hz]. Each plot shows the phase-coupling between the reference contact and all other contacts (−750–1200 μm) for the sustained response 200–500ms after stimulus onset. The color scaling (amplitude of the averaged LFPbp) is the same for all plots within a row. Blue color indicates a trough, red colors indicate a peak in the broadband LFPbp.

The online version of this article includes the following figure supplement(s) for figure 6:

**Figure supplement 1.** Phase-triggered-averages (PTA) of local LFPbp aligned to six different reference channels (top to bottom: 750, 450, 150,−150, −450 μm) for the $\theta$ frequency band [4–8 Hz], The figure follows the conventions set out by *Figure 6* in the main text.

**Figure supplement 2.** Phase-triggered-averages (PTA) of local LFPbp aligned to six different reference channels (top to bottom: 750, 450, 150, −150, −450 μm) for the α frequency band [8–13 Hz], The figure follows the conventions set out by *Figure 6* in the main text.

*Figure 6 continued on next page*

*Figure 6 continued*

**Figure supplement 3.** Phase-triggered-averages (PTA) of local LFPbp aligned to six different reference channels (top to bottom: 750, 450, 150,–150, –450 µm) for the β frequency band [14–25 Hz], The figure follows the conventions set out by *Figure 6* in the main text.

**Figure supplement 4.** Phase-triggered-averages (PTA) of local LFPbp aligned to six different reference channels (top to bottom: 750, 450, 150,–150, –450 µm) for the γ frequency band [35–55 Hz], The figure follows the conventions set out by *Figure 6* in the main text.

**Figure supplement 5.** Phase-triggered-averages (PTA) for comparison to Figure 3 in *van Kerkoerle et al., 2014*.

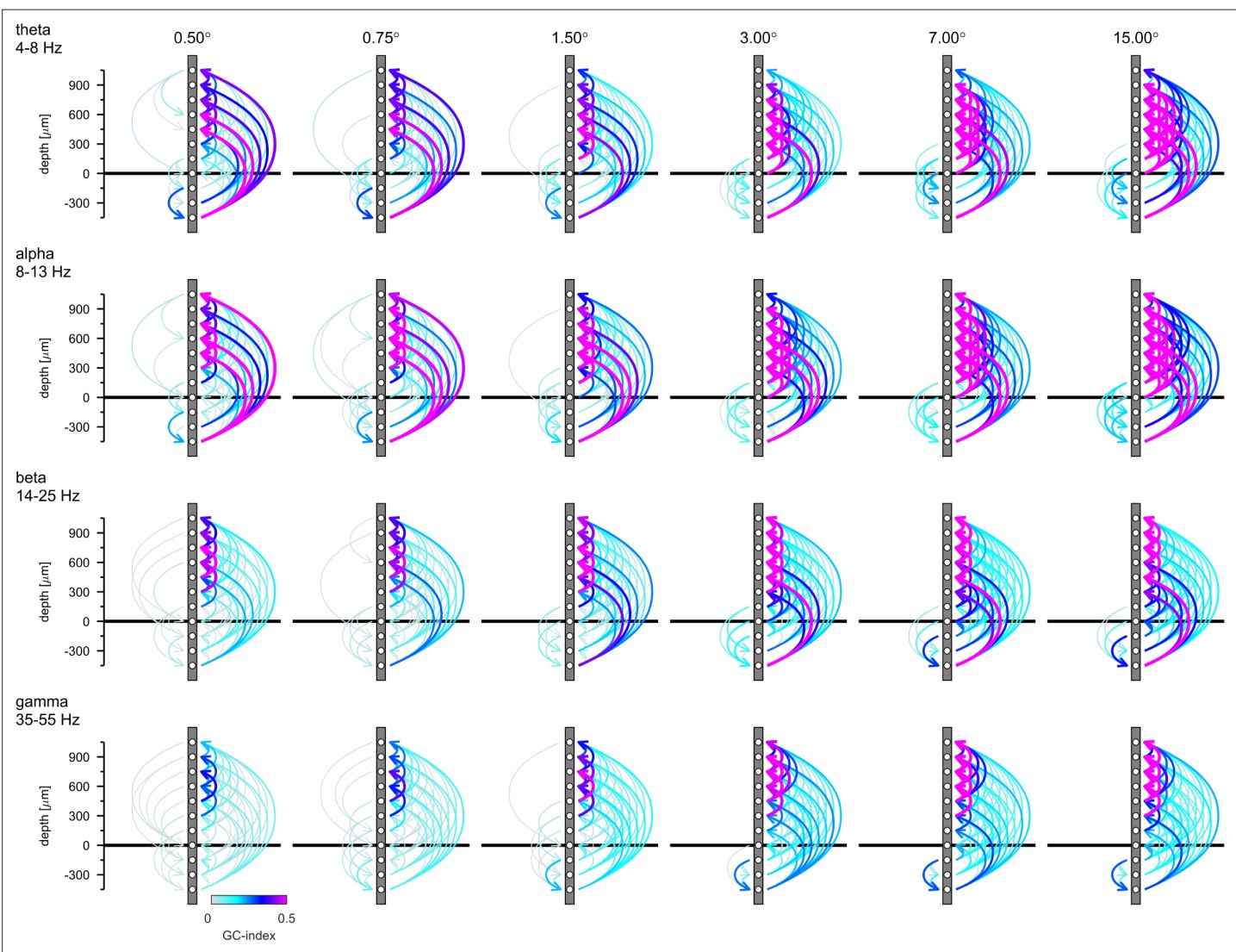

**Figure 7.** Granger-Causality (GC) analysis reveals directed coupling between cortical layers. Causal-coupling pattern derived from non-conditional GC analysis in the frequency domain and tested with time-reversed GC analysis. GC-indices were integrated across the four classic frequency bands (top row: θ [4–8 Hz], second row: α [8–13 Hz], third row: β [14–25 Hz], bottom row: γ [35–55 Hz]). Each arrow indicates the dominant direction of causality between a given pair of contacts. The color represents the value of the difference between the GC-indices in each direction. Downward directions are plotted on the left, upward directions are plotted on the right of each profile.

The online version of this article includes the following figure supplement(s) for figure 7:

**Figure supplement 1.** Spectral Granger-causality for selected connections between infragranular and supragranular layers.

**Figure supplement 2.** Connection plots of net Granger-causality in four frequency bands.

**Figure supplement 3.** Connection plots of net Reverse Granger-causality in four frequency bands.

**Figure supplement 4.** Comparison of Granger-causality indices for LFPbip and MUAe.

In the upward direction strong θ-GCs originated in layer 6 (lower contacts) affecting predominantly supragranular layers, irrespective of stimulus size. With increasing stimulus size GCs increased in an upward direction between granular and supragranular, and between supragranular-supragranular contacts. Overall, a very similar pattern was present for GCs in the α- frequency band. In the β-frequency band, GCs were fairly weak in the downward direction, when small stimuli were presented, but increased between layer 5 and layer 6 for large stimuli. Upward GCs were pronounced between granular and supragranular, and between supra-supragranular contacts for all stimulus sizes. With increase in stimulus size a pronounced infragranular-supragranular GC occurred. An overall similar pattern was present for γ-frequency GCs, with an even stronger increase in upwards directed supra-supragranular GCs with stimulus size. The results from *Figure 7* show that surround modulation results in strongly increased low, mid and high frequency band communication between granular and supra-granular compartments, and it results in strongly increased γ-band communication originating from layer 5, which drives γ frequency changes in layer 6, and which in turn changes γ frequency oscillations in upper layer four and in layers 2/3. Performing a similar analysis using MUAe signals instead of LFP signals results in overall similar pattern, with an additional component of γ frequency GC originating in layer 4 (*Figure 7—figure supplement 4*).

## Discussion

Using stimuli of different sizes in combination with laminar recordings in primary visual cortex of the macaque revealed that narrow band γ frequency oscillations occur across cortical layers and increase across cortical layers with increasing stimulus size. When stimuli encroached on the far surround of the receptive field, the LFP γ oscillations in layer 5, the feedback recipient layer, exerted pronounced granger causal influences on γ oscillations in layer 6, which in turn granger causal influenced γ oscillations in supragranular layers. This pattern of γ frequency granger causal interactions is compatible with the known anatomical termination sites of feedback connections into V1 (*Rockland and Pandya, 1979*). The flow of information (granger causal interactions in the γ frequency range) occurred in a manner compatible with the canonical microcircuit, but in a largely reverse direction. Here, information terminating in layer 5, is passed on to layers 6 and 4, who in turn send the information to layers 2/3. This demonstrates that γ oscillations are neither restricted to superficial layers, nor are they invariably signatures of feedforward communication. Moreover, low-frequency oscillation GC interactions did not strictly adhere to their proposed role in feedback communication, showing most profound increases in granular to supragranular and supra- to supragranular interactions with stimulus size. Feedback influence terminating in layer five was not dominated by low-frequency oscillations as predicted by recent publications (*Bastos et al., 2015a*; *Bastos et al., 2015b*; *Bosman et al., 2012*; *Buffalo et al., 2011*; Buschman and Miller; *van Kerkoerle et al., 2014*), even though GC interactions in low-frequency bands in the intracolumnar feedback direction (infragranular to granular/supragranular layers) were profound (*Figure 7—figure supplement 1*). This demonstrates that low-frequency oscillations are not restricted (or dominant) to infragranular layer and can attain feedforward roles in V1. As described in Materials and methods, the stimuli were optimized in relation to orientation tuning in one of the monkeys, but not the other. However, the key results reported in the paper did not differ between animals, which suggests that stimulus optimization is not critical for these results to be obtained.

The differences to previous publications could be due to the use of cognitively demanding task engagement (*Bastos et al., 2015a*; *Bastos et al., 2015b*; *Bosman et al., 2012*; *Buffalo et al., 2011*; *Buschman and Miller, 2007*; *van Kerkoerle et al., 2014*) vs. passive fixation as used in our task. Additionally, the use of large grating stimuli might be a contributing factor, as our phase lag and GC results using small stimuli are more similar to those reported in other publications (*van Kerkoerle et al., 2014*), even though their stimuli are not directly comparable to our small grating stimuli.

### Response latencies and surround suppression as a function of layer and stimulus size

In line with predictions from the anatomy (*Callaway, 1998*; *Callaway, 2004*; *Lund, 1988*), the earliest responses in V1 occurred in layer 4 c, aligned with the earliest current sink. This was followed by responses in layer 6. Responses in supragranular layers were delayed by up to 20ms. These results

qualitatively match previous reports in the anesthetized macaque (*Nowak et al., 1995*) and awake monkeys (*Self et al., 2013*). The early responses in layer six are likely mediated by direct inputs from the LGN (*Callaway, 1998*; *Callaway, 2004*; *Lund, 1988*). The relatively faster responses in layer 5 (relative to layer 2/3) are somewhat more surprising, as connections from layer 4c to layer 5 are less dense than those to layer 2/3 (*Callaway, 1998*). However, the connections to layer 2/3 possibly involve more synapses. Stimulus size affected response latencies in layer 4, where increasing stimulus size initially resulted in decreased latencies, probably due to increased excitatory drive, followed by increased latencies when stimuli were larger than 1.5°. This increase is most likely driven by surround suppression at the level of the LGN and the retina (*Alitto and Usrey, 2008*; *Bijanzadeh et al., 2018*). In layers 5 and 6, increasing stimulus size resulted in shorter response latencies with an asymptote reached once stimuli were 1.5° in size. This could be due to a mixture of increased excitation from LGN inputs, and fast feedback inputs (*Bair et al., 2002*; *Bijanzadeh et al., 2018*) that compensate for the surround suppression inherited from feedforward inputs when stimuli exceed 1.5°.

Surround suppression equally differed between cortical layers. Surround suppression was largely homogenous for supragranular and granular recording sites, but decreased notably for layers 5 and 6. Both findings are equivalent to previous investigations in the anaesthetized monkey (*Sceniak et al., 2001*). We speculate that the decreased surround suppression in layers 5 and 6 is a consequence of the added feedback drive (*Angelucci et al., 2002*; *Bair et al., 2002*; *Bijanzadeh et al., 2018*; *Rockland and Pandya, 1979*) from higher cortical areas.

## Gamma oscillations for different stimulus sizes and layers

In line with previous studies, we show that the strength of narrow band γ frequency oscillations increased monotonically with stimulus size (*Bauer et al., 1995*; *Gieselmann and Thiele, 2008*; *Jia et al., 2011*; *Peter et al., 2019*; *Ray and Maunsell, 2010*). γ frequency power was largely broadband for small stimuli, and most profound in supragranular layers. With increasing stimulus size, γ frequency power became more narrowband with peak frequencies at ~40–50 Hz. With increasing size, power relative to baseline was relatively strong across all cortical layers, including layer 4, even if it was most profound in supra- and infragranular layers. These findings somewhat differ from previous reports where γ frequency power were much stronger in supragranular than infragranular layers (*Bollimunta et al., 2011*; *Maier et al., 2010*; *Spaak et al., 2012*; *Xing et al., 2012*). However, some studies have reported increased γ frequency power in granular and infragranular layers upon stimulus onset (*Maier et al., 2011*; *van Kerkoerle et al., 2014*). Neither of these have evaluated the effect of stimulus size on γ frequency power across layers. *van Kerkoerle et al., 2014* reported that an increase in α frequency power upon structured background presentation in infragranular layers is a dominant feature of the spectral pattern of LFP activity. This increase was relative to the pre-stimulus period, as well as to the condition when a figure was presented in the neurons' receptive field. However, they also saw changes in the γ frequency, which were stronger for figure conditions, but also present for background conditions. The latter changes were most pronounced in supragranular layer and layer 6, roughly in line with our results.

We found that increasing stimulus size, which invokes strong feedback signals terminating in layers 1 and 5 (*Bijanzadeh et al., 2018*) resulted in strongly increased narrow band γ power in layer 5. These in turn Granger causal affected γ frequency power in layer 6, from where γ frequency power in layers 4, 3, and 2 were affected. This shows that γ frequency oscillations are not invariably a signature of feedforward processing (*Bastos et al., 2015a*; *Bosman et al., 2012*; *Roberts et al., 2013*; *van Kerkoerle et al., 2014*), but can be an important signature of feedback processing. γ gamma frequency oscillations can be profound in V2 (*Frien et al., 1994*; *Roberts et al., 2013*), and cooling of V2 reduces γ gamma frequency oscillations in V1 (*Hartmann et al., 2019*). Thus, we speculate that the γ gamma frequency oscillations sources in layer five are inherited from downstream areas through feedback connection. Which layer these originate from is unknown. They could originate from supragranular sources (*Markov et al., 2014*) of hierarchically close downstream areas, such as V2 or V3 (*Markov et al., 2014*). However, if there were a cascade of feedback projections from downstream areas (e.g. V4> V2, V2> V1), each strongly oscillating in the γ frequency range, then even V2 might show the pattern we describe for V1 here, that is strong γ frequency in infragranular layers. This entails the possibility that the γ frequency feedback could originate from V2 infra- and supragranular layers.

At the same time, it has been shown that feedback from hierarchically close areas (such as V2-> V1) can predominantly arise in layer three and target layer 3 (*Markov et al., 2013*), even though feedback from hierarchically close areas always has a component targeting deep layers (*Barone et al., 2000*). This would predict that stimulus-size-induced changes in granger causal interactions would predominantly occur in layer three if they were triggered from area V2, a pattern not seen in our data. Long-range feedback projections on the other hand terminate predominantly in layer 1 and layer 5 (*Markov et al., 2013*; *Rockland and Pandya, 1979*). Given the phase relationship between γ frequency oscillations in different layers and the changes in Granger causal interactions originating in layer 5 with large stimuli, we suggest that some component of stimulus-size-induced changes in oscillatory activity arises from longer range feedback connections.

Feedback invoked by stimulus manipulations (i.e. an automatic 'bottom-up' type of feedback) could arise pre-dominantly in hierarchically close areas (e.g. V2), and might be less pronounced from hierarchically distant areas, where lower frequency would dominate due to their infragranular sources (*Buffalo et al., 2011*; *Markov et al., 2014*). It is possible that this differs from more cognitively driven 'top-down' feedback signatures, which could in principle use a different (lower) frequency range, due to the longer range interactions necessary (*Bastos et al., 2015a*; *Bosman et al., 2012*; *Buffalo et al., 2011*; *van Kerkoerle et al., 2014*). This could explain the differences between our results and those described in *van Kerkoerle et al., 2014*, who engaged their animals in a figure ground detection task and an attention demanding curve tracing task. Contrary to *van Kerkoerle et al., 2014*, we also did not find that α frequency, or other low frequency oscillations, bore the hallmark of feedback signals. They were not restricted to infragranular layers or layer 1, and α frequency-based communication within V1, while strongly progressing from layer 6 to supragranular layers for all stimulus sizes, attained prominence within granular and supragranular layers for large stimulus sizes and was directed in an upward direction from there. These results shows that different frequency bands show hallmarks of feedforward and feedback communication in a flexible, stimulus dependent manner.

We propose that 'cognitive signal' feedback operates between retinotopically aligned neurons (receptive field center aligned), while bottom-up driven feedback operates between retinotopically non-aligned neurons (receptive field center offset). The former could be predominantly excitatory in nature, while the latter might result in net inhibition (even if indirectly). This would explain, why cooling higher areas results in reduced responses in V1 to stimuli restricted to the classical receptive field, but increased responses to stimuli extending into the receptive field surround (*Hupé et al., 1998*). To clarify whether an automatically (stimulus driven) triggered type of feedback really differs from a 'cognitive' top-down feedback signal, it will be necessary to investigate the layer dependency and the frequency content of interareal interactions under passive viewing conditions and under conditions of active stimulus selection (e.g. spatial attention) demands in future experiments.

## Materials and methods

Two awake male rhesus monkeys were used for the electrophysiological recordings reported in this study. After initial training, monkeys were implanted with a head holder and recording chambers above V1 under general anesthesia and sterile conditions (for details of surgical procedures, postsurgical analgesics, and general postsurgical treatment, see *Thiele et al., 2006*). All procedures complied with the European Communities Council Directive RL 2010/63/EC, the U.S. National Institutes of Health Guidelines for the Care and Use of Animals for Experimental Procedures, and the UK Animals Scientific Procedures Act. Animals were motivated to engage in the task through fluid control at levels that do not affect animal physiology and have minimal impact on psychological wellbeing (*Gray et al., 2016*).

### Data acquisition

Recordings were performed using a 16-contact microelectrode (V-probe, Plexon, Dallas, Tex., US). Contacts were arranged vertically along the electrode with an inter-contact spacing of 150 µm. Probes were lowered into occipital cortex through the intact dura mater by a manually operated hydraulic microdrive system (Narishige International ltd, Japan). Extracellular voltage fluctuations were amplified by means of the Digital Lynx recording system (Neuralynx, Bozeman, Mont., US). The raw signal of each electrode contact was referenced to the guide tube and recorded with 24 bits resolution at a

sampling rate of 32 kHz. This raw signal was then processed offline to produce two types of signals, the local field potential (LFP) and the envelope of multi-unit activity (MUAe). It is assumed that the LFP reflects the low-frequency changes of post-synaptic potentials (*Mitzdorf, 1987*), while MUAe captures the aggregate high-frequency spiking activity of the population of neurons around the electrode contact (*Supèr and Roelfsema, 2005*). The LFP was obtained by low-pass filtering (IIR-Butterworth, 3rd order, 0.75–300 Hz) and down-sampling the raw signal to ~1 kHz. To obtain MUAe, we applied a high-pass filter (IIR-Butterworth, 3rd order, 0.6–9 kHz) and full-wave rectified the raw signal before low-pass filtering at 200 Hz and down-sampling to ~1 kHz (*Supèr and Roelfsema, 2005*).

## Task and visual stimulation

Experimental timing, behavioral control and stimulus presentation was controlled by Cortex (DOS-Version 5.95; NIMH) running on IBM-compatible personal computers. The subjects were seated in a primate chair and performed a passive fixation task. A subject started a trial by directing its gaze toward a small fixation spot (red annulus, diameter 1.6°) on a CRT-monitor (monkey 1: 1024 × 768 at 120 Hz; monkey 2: 1280 × 1024 at 100 Hz) and received a juice-reward after keeping fixation for another 3–4 s. In each trial we presented a sequence of 3 or 4 (monkey 1 or monkey two respectively) grating stimuli centered on the receptive field (RF) of the neuronal column under study.

Each grating was presented for 500ms, preceded by a blank period of 500ms. Gratings were square-wave modulated (duty-cycle = 1.5/°) and varied in size between six different diameters (0.5°, 0.75°, 1.5°, 3.0°, 7.0°, 15.0°). In monkey one the orientation of all gratings was adjusted to the preferred orientation of the neuronal column under study. In monkey 2, we also varied the orientation of the grating between 12 equally spaced orientations.

## Experimental protocol

When lowering the electrode, we visually monitored the activity recorded from the probe by means of the recording software. We were able to distinguish cortical LFP activity from the signal recorded above the pial surface by signal amplitude, frequency content and the presence of visually evoked potentials (VEP). In each experiment we initially lowered the electrode to a point in depth where we could identify cortical LFP in the lower 10–12 channels of the probe. We then left the probe in place for a minimum resting period of 30 min to achieve stable recording conditions. During this resting period cortical LFP activity was usually travelling upwards along the channel array of the probe as a consequence of stress relief on the cortical tissue caused by the penetration. After the resting period we finally adjusted the depth position of the probe such that the two uppermost channels of the array did not show any clear VEPs.

We then started estimating the RFs of the cortical column by using a reverse-correlation technique (described in *Gieselmann and Thiele, 2008*). In short, the RF was mapped by presenting a dark gray square (0.25° x 0.25°) changing positions randomly drawn from a 12 by 9 grid of possible positions every 130ms. The experimenter then estimated the RF center from the spike probability at each of the grid position. For monkey one, we estimated the preferred orientation in a similar fashion (described in *Gieselmann and Thiele, 2008*).

## Extracting the locally generated LFP

The LFP is a sum of a number of locally generated components which are shared across multiple neighboring electrodes due to volume conduction (*Mitzdorf and Singer, 1979*; *Schroeder et al., 1991*). To extract the locally generated LFP we transformed the LFP in two ways depending on the two response periods of interest.

The early, 'phasic' part of the neuronal response (0–200ms after stimulus onset) contains the VEP, which are transient voltage deflections in the LFP time-locked to the stimulus onset. To obtain the VEP, we first averaged the LFP-response in each of the 16-channels over all available trials, eliminating all components that are not phase-locked to stimulus onset. Then we computed the CSD from the trial-averages of each channel using the 'standard' method of the inverse-CSD (iCSD) toolbox (*Pettersen et al., 2006*). iCSD is computed as the second spatial derivative of a particular electrode and its neighbors (*Freeman and Nicholson, 1975*; *Mitzdorf and Singer, 1979*; *Schroeder et al., 1991*):

$$CSD = -\sigma \frac{\phi(z+h) - 2\phi(z) + \Phi(z-h)}{h^2}$$

where σ is the extracellular conductivity, Φ the potential (VEP), h the interelectrode spacing and z the depth of the electrode contact under study. We used a value of σ = 0.3 S/m for the extracellular conductivity. This method itself discards the top and bottom channel. To regain the top and bottom channels we applied the method described by Vaknin (*Pettersen et al., 2006*; *Vaknin et al., 1988*).

The late, 'tonic' part of the neuronal response (200–500ms after stimulus onset) contains components that are not time-locked to the stimulus onset but are generated by the dynamics of the neuronal network. To extract the locally generated components at a given contact we computed the bipolar LFP (LFPbp) as the difference between the LFP at its neighboring contacts for every trial to minimize common signals. The LFPbp is less prone to noise induced spurious results when performing connectivity analysis like Granger causality (*Trongnetrpunya et al., 2016*). Subsequently, the LFPbp was subjected to spectral analysis methods.

## Electrode perpendicularity and intersession alignment

To interpret the laminar specificity of neuronal activity, each of the 16 channels had to be assigned a putative position within the cortical depth for every session (*Figure 1*). The position in depth varied across sessions due to manual advancement of the probe and the compressions-relaxation dynamics of the cortical tissue during the penetration. For early visual cortex, several studies compared the visually evoked activity with the laminar anatomy after histological reconstruction of recording sites (*Kraut et al., 1985*; *Mitzdorf and Singer, 1979*; *Schroeder et al., 1991*). These studies established that an early (40ms after stimulus onset) current sink in the CSD-profile corresponds to the thalamo-cortical afferent activity reaching layer 4 c. This alignment-criterion has been successfully applied in a number of recent studies of primary visual cortex V1 using laminar electrodes (*Maier et al., 2010*; *Maier et al., 2011*; *Self et al., 2013*; *Spaak et al., 2012*; *van Kerkoerle et al., 2014*). To align electrode position across sessions, we computed the CSD-profile across all stimulus presentations of each session and visually identified the channel that featured an early current sink, peaking between 35 and 55ms after stimulus onset. *Figure 1* illustrates the alignment procedure applied to our data.

## Data analysis

All data analysis was performed with custom-written code in the MATLAB computing environment (The Mathworks, Natick, Mass., USA).

The latency of MUAe responses in each layer was determined by analyzing a model of the visually evoked transient from 0 to 200ms after stimulus onset. The model was computed by a fitting procedure adapted from *Self et al., 2013* which fitted the sum of two Gaussians and a cumulative Gaussian by a non-linear least-squares fitting procedure (MATLAB, Curve Fitting Toolbox):

$$R(t) = \frac{G_1 e^{-0.5\left(\frac{t-\mu_1}{\sigma_1}\right)^2}}{\sqrt{2\pi}\sigma_1} + \frac{G_2 e^{-0.5\left(\frac{t-\mu_2}{\sigma_2}\right)^2}}{\sqrt{2\pi}\sigma_2} + 0.5G_3 \left[1 + erf\left(\frac{t-\mu_3}{\sqrt{2\sigma_3^2}}\right)\right]$$

R(t) is the response at time t defined by the parameters of the three Gaussians, (G: amplitude, μ: mean, σ: standard deviation) and erf is the error function. The function was fitted separately to the responses at each electrode contact to each of the six stimulus sizes. Before the fitting procedure the MUAe signal was z-scored relative to the spontaneous activity 0–300ms before stimulus onset. The stimulus size is negatively correlated to the amplitude of visual responses due to surround suppression. Hence, different from *Self et al., 2013*, we used an absolute criterion to determine the response latency as we sought to compare responses to different stimulus sizes. The latency was defined as the earliest time the model exceeded a z-score of 3.

For general spectral analysis of the LFP we computed the spectrogram by applying fast Fourier transformation (FFT, n = 1024) to data sections from a moving window (width = 100ms, step = 10ms). Each data section was 'Hanning-windowed' before FFT was applied.

We computed the LFP phase-triggered average (PTA) to estimate the phase-coupling of oscillations between electrode contacts in the θ (4–8 Hz), α (8–13 Hz), β (14–25 Hz), and γ (30–55 Hz) frequency bands. First, for each frequency band, we extracted the phase providing signal ($S_{phase}$) from a reference channel by band-pass filtering (butterworth filter, order: n = 3, cutoff frequency: $\theta$ = 4–8, $\alpha$ = 8–13, $\beta$ = 14–25, $\gamma$ = 30–55 Hz). Then, the broadband signal of the amplitude providing signal ($S_{amp}$)

was re-aligned to the troughs of $S_{phase}$ and averaged within±120, 80, 80 or 25ms (θ, α, β, or γ) of each alignment point to yield the PTA-histogram.

The direction of coupling between the LFPbp of different channels was analyzed by means of non-conditional granger-causality-analysis (GC). Here, we used the MVGC-toolbox, which is freely available on the web (*Barnett and Seth, 2014*). We computed granger-causality indices (GCI) for a stretch of 256 time bins from 200ms after stimulus onset on. For the multivariate autoregressive model, we used a fixed model order of 50 (~50ms at 1017 Hz). To extract causal unidirectionality between a certain channel pair we subtracted GCIs calculated in one direction from the GCIs in the reverse direction. We computed GCIs for all channel combinations within an experiment.

## Acknowledgements

We thank the Comparative Biology Centre staff at Newcastle University for their excellent technical support.

## Additional information

### Funding

| Funder | Grant reference number | Author |
| --- | --- | --- |
| Medical Research Council | MR/P013031/1 | Marc Alwin Gieselmann Alexander Thiele |
| The Wellcome Trust | 093104/Z/10/Z | Marc Alwin Gieselmann Alexander Thiele |

The funders had no role in study design, data collection and interpretation, or the decision to submit the work for publication.

### Author contributions

Marc Alwin Gieselmann, Conceptualization, Data curation, Formal analysis, Investigation, Methodology, Software, Validation, Visualization, Writing – review and editing; Alexander Thiele, Conceptualization, Funding acquisition, Project administration, Resources, Supervision, Writing - original draft, Writing – review and editing

### Author ORCIDs

Marc Alwin Gieselmann http://orcid.org/0000-0002-8034-5134
Alexander Thiele http://orcid.org/0000-0003-4894-0213

### Ethics

All procedures complied with the European Communities Council Directive RL 2010/63/EC, the U.S. National Institutes of Health Guidelines for the Care and Use of Animals for Experimental Procedures, and the UK Animals Scientific Procedures Act and were reviewed and approved by the University animal care, welfare and ethical review body (AWERB). The animals were group housed together in groups of between two and three animals in both same and mixed gender groups. The housing and husbandry complied with the guidelines of the European Directive (2010/63/EU) for the care and use of laboratory animals and following the Animal Research Reporting of In Vivo Experiments (ARRIVE) principles on reporting animal research. All surgeries were performed under sevoflurane anaesthesia. Animals received post-surgical analgesia treatment as deemed necessary by the veterinary team.

### Decision letter and Author response

Decision letter https://doi.org/10.7554/eLife.62949.sa1
Author response https://doi.org/10.7554/eLife.62949.sa2

## Additional files

### Supplementary files
• Transparent reporting form

### Data availability
Original source data from all relevant data sets are available at DOI: https://doi.org/10.25405/data.ncl.17136182.v1 in the data depository of Newcastle University (https://data.ncl.ac.uk/ managed by figshare). Data are in binary format (Neuralynx) and are in the form of original recordings, along with relevant event files.

The following dataset was generated:

| Author(s) | Year | Dataset title | Dataset URL | Database and Identifier |
|---|---|---|---|---|
| Gieselmann M, Thiele A | 2022 | Stimulus dependence of directed information exchange between cortical layers in macaque V1 | https://doi.org/10.25405/data.ncl.17136182.v1 | Newcastle University, 10.25405/data.ncl.17136182.v1 |

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
