## [Editor Report]

The cortical column can be considered a mesoscopic scale unit of computation. While neural anatomy suggests a well-known flow of information from granular to supra-granular to infra-granular layers, how well this maps onto feedforward and feedback signal flow in different frequency bands is less clear. In this paper Gieselmann and Thiele show that stimulus properties play a crucial role in determining the flow.

---

## [Decision Letter]

**Decision letter after peer review:**

Thank you for submitting your article "Stimulus dependence of directed information exchange between cortical layers in macaque V1" for consideration by *eLife*. Your article has been reviewed by 2 peer reviewers, and the evaluation has been overseen by a Reviewing Editor and Joshua Gold as the Senior Editor. The following individual involved in review of your submission has agreed to reveal their identity: Scott Lowe (Reviewer #1).

Below you will find a set of recommendations, which, if completed, would lead to publication in *eLife*.

Essential revisions:

1) GC Analysis using MUA (see below, rev1.1).

2) A more extensive of the distribution of receptive field sizes, and if appropriate, repeating the analysis when using 1 deg as reference (rev 1.2, editor comments).

3) Clarify text regarding the difference between lateral intracolumnar feedback and top-down intra-area feedback (rev 2).*Reviewer #1:*

In this work, the authors study neural oscillations induced in macaque V1 by square-wave stimuli of varying sizes. The results suggest that α/β frequencys bands arise in L6, with long-range interactions to L2/3; and γ frequency arise in L4 or lower L3, with a series of short range interactions up the cortical column, all the way to L1. There is also an interaction from ~L4C to L6, for which the frequency changes depending on the size of the stimulus (low frequency for small and high frequency for large stimuli).

I find the paper is worthy of publication in *eLife* as it stands, and could be made stronger with some additional work.

1. The GC analysis was applied to four frequency-bands: theta, α, β, and low-γ. The same analysis could easily be applied to the MUA. Also, there is the possiblity of cross-frequency coupling (Spaak et al., 2012): high-frequency oscillations in one region may induce low-frequency oscillations in another (and vice-versa), and this could be studied with GC between different frequencies. Why is such analysis excluded from the paper? It would seem to be a natural fit, and I think this additional work would make the paper stronger.

2. Much of the discussion throughout the paper focuses on the difference of effects with small centre-only stimuli and large stimuli which stimulate the surround. However, the distribution of RF size of the cortical columns being studied is not reported. This should be included so the size of the stimuli can be contextualised. I also posit that correcting for the size of the RF might lead to a clearer transition from the centre to surround recording condition. If there is a large range of RF sizes between sessions (a factor of 2 difference, or more) I ask that the authors consider this, or explain why correcting for the RF size would not be desirable.

3. On L160, the authors cite Einevoll to justify using the LFPbp, but I am sure he would instead advocate to use the CSD instead. Why use the LFPbp instead of the CSD?

4. There are more accurate methods of estimating the CSD than the double-spatial derivative. There is the inverse-CSD (iCSD; Pettersen, 2006) and kernel-CSD (kCSD; Potworowski, 2012). Problems can arise with the double-spatial derivative when the current sources are small compared to the electrode measurement interval. These new methods take into account the electrode sampling interval, and the expected spatial extent of sources, in order to estimate the CSD from the LFP. Why use the double-spatial derivative instead of iCSD or kCSD?

5. With regards to determining the spectral power (L483), why not use the Welch method, or similar, instead of just the FFT? This would give a more stable estimate of power, especially in the high frequencies.

6. It is curious that you find stimulus-driven changes centred at a frequency of 40Hz (Figure 4). Earlier works have found that stimulus-driven information is isolated into two compartments, one below 40Hz and the other above 40Hz (Belitski, 2008) and that frequencies below and above 40Hz encode information about different spatio-temporal components of stimuli (Lowe, 2017), whereas (in these works) frequencies around 40Hz do not contain information about the stimulus. Furthermore, they appear to be driven by dopaminergic neuromodulation (Zaldivar, 2018). Also, it is noteworthy that the stimulus presentation spatial frequency of 1.5cpd is the point where γ frequencies (<40Hz) become more informative about the stimulus than theta and α (Lowe, 2017).

7. I am curious as to what other CSD-profiles were seen (L72). Were they just noisy, or indications that the probe was not aligned with cortical column? Or is there some (albeit weak) evidence CSD profiles are not uniform across V1? I assume the former, but ask in case there is a genuine minority CSD profile which multiple labs are seeing and rejecting. I have rejected a session before which was perfect, except for the inconvenient fact that the sign of all CSD currents were inverted.

Refs:

Belitski et al., (2008). doi:10.1523/JNEUROSCI.0009-08.2008.

Logothetis et al., (2007). doi:10.1016/j.neuron.2007.07.027.

Lowe (2017). https://ethos.bl.uk/OrderDetails.do?uin=uk.bl.ethos.738787

Pettersen et al., (2006). doi:10.1016/j.jneumeth.2005.12.005.

Potworowski et al., (2012). doi:10.1162/NECO_a_00236.

Zaldivar et al., (2018). doi: 10.1016/j.cub.2017.12.006.*Reviewer #2:*

Anatomical tracing studies have established that the connectivity between visual areas abides by a global hierarchy, such that between any two visual areas, there is a graded hierarchical relation. Prior studies (Bastos, Neuron, 2015; Michalareas, Neuron, 2016) have shown that α and β GC is stronger in the top-down direction, whereas theta and γ GC is stronger in the bottom-up direction. Within a cortical column, the concept of a canonical microcircuit suggests a flow of information from granular to supragranular and then to infragranular layers. Prior studies (Livingstone, J. Neurophys., 1996; van Kerkoerle, PNAS, 2014) have shown that γ oscillations have characteristic laminar phase relations, with systematic lags as function of laminar distance from granular, and that α oscillations can show the opposite, i.e. systematic leads. These two sets of results, i.e. interareal and intracolumnar, are consistent, but need to be regarded separately. The hierarchical ordering of visual areas is supported by unequivocal evidence, whereas a columnar information-flow pattern is merely suggested, and different models are consistent with the empirical evidence. The current manuscript speaks to the columnar level and presents interesting evidence, partly consistent with van Kerkoerle, and partly extending the van Kerkoerle study to different stimulus sizes. It does not speak directly to the interareal level. The authors need to revise their manuscript to avoid confusion.

Figure 6 and related figures and text: This is almost the same analysis as performed by van Kerkoerle, with the important difference that Kerkoerle used CSD, whereas the present manuscript uses bipolar LFP. The present manuscript makes direct comparisons to Kerkoerle and partly reports conflicting results. For a sound and fair comparison, the authors of the current manuscript need to provide these analysis based on the CSD.

The phase-lag and GC results are reported as mainly contradicting previous results. However, if only the results are considered that use exactly the same methods as previous studies (e.g. Kerkoerle with the same stimulus size), results are not so differet. Results obtained with other stimulus sizes are not in conflict with this. And all results of those intra-V1 recordings are not in conflict with interareal GC analyses. Thus, the authors should limit reported conflicts to aspects that can really be compared.

The same holds for the reporting of power. For example line 307 – 309: Xing, 2012 did show γ power in many layers, with a distinct peak in deep L5/upper L6; and Buffalo, 2011 focused on spike-field coherence. SFC is a very different metric from LFP power. It is possible that a particular layer, for a particular stimulus size, shows clear γ LFP power, but maybe the spikes there do not phase-lock to the LFP γ phase, such that γ SFC is low. Thus, again, the authors should limit reported conflicts to aspects that can really be compared.

I am somewhat concerned about the upwards dominance of GC (Figure 7 supp 1). Could there be any trivial reason for this?

Line 326: Early work from the Eckhorn lab showed clear γ in V2. A recent preprint from the Richard Born lab shows strong reductions in V1 γ upon V2 cooling. Also, Bastos, Neuron, 2015 showed clear V2-to-V1 γ GC; note that this paper relates the DAI to the SLN in a graded manner, and the known feedforward-type (supragranular originating) projections from V2 to V1 are expected to carry γ in this top-down direction. Thus, we do know that the input feedback signal shows γ.

Figure 4D: It would be very helpful to see spectra (cross-cuts through the presented spectro-depth plots), at the depths with strongest γ, i.e. around 450 and -450

Figure 7 and similar plots: This is a matter of taste, but I find a matrix version of such results much easier to parse.

Line 37,38: Interareal theta-band GC is stronger in the feedforward, not the feedback, direction (Bastos, Neuron, 2015; Spyropoulos, PNAS, 2018). The feedforward nature of theta is actually correctly referred to in line 144.

Line 44: Classical work by Lopes da Silva located the α source rather deep (Lopes da Silva, Neuroscience Letters, 1977, pages 237 – 241).

Lines 69-74: Clarify whether all three conditions have to be met, or whether one suffices.

Line 104: How was the ANOVA done? Is it suitable to provide the p-value for the statement in the preceding sentence on 3 ms longer latencies?

Line 107: The figure lists p=0.98, whereas the text lists p=0.59. Also, the text specifies a one-way ANOVA. Please clarify.

Line 169: Gieselmann and Thiele previously showed an actual γ peak for small stimuli, whose peak frequency decreased with stimulus size. This should be related to the current results.

Figure 4B, D and Figure 5A, D, G, J: The text and figure legends are quite clear on that these plots show power changes due to stimulation. I am surprised that changes are exclusively positive. Previous studies have found stimulus induced power decreases, e.g. in α. Are these rectified power changes? In any case, the authors should address this excplicitly, simply to avoid confusion. Also, where the text refers to those panels, it should be careful to actually refer to power changes (not just power).

Line 177 and a few other places: the Xing, Shapley, PNAS, 2012 paper showed several γ peaks along cortical depth, including a weak one at lower layer 5/upper layer 6.

Line 193: I was surprised by the use of the term phase-amplitude-coupling for this analysis. Usually, PAC is used for the relation of low-frequency phase to higher-frequency power, not the amplitude of the raw signal.

Line 220-221: This statement is true for one isolated phase lag value. However, a pattern of phase lags as analyzed by Kerkoerle or in the present manuscript (or in the context of phase-lag spectra) resolves these uncertainties. However, the statement can be made that phase relations are sometimes not consistent with GC (explicitly mentioned by Brovelli, PNAS, 2002, start of page 9853), and this can be resolved with GC methods (Witham, Riddle, Baker, J Physiol, 2011).

Line 264: Figure 7 supp 1 B shows the dominant influence of infra onto supra to be around 20 Hz, i.e. in the same β band that Bastos, Neuron, 2015 found to be feedback related. Figure 7 supp 1 G shows infra to very supra and shows a broad α peak that is at least compatible with Buffalo, PNAS, 2011.

Line 301, 302: Also Peter, Fries, Vinck, *eLife*, 2019.

Line 335: Barone, JNS, 2000 shows an SLN of 47 %, i.e. not predominantly supragranular, and most likely not predominantly from just layer 3.

Line 339-342: Top-down projections, even if short-range, always have a component of feedback-type connectivity, targeting deep layers. Thus, no reason to invoke longer-range feedback connections.

Line 483: How do the authors obtain 1000 samples in 100 ms? Are they transforming a 10 kHz signal? If all FFT analys were based on 100 ms windows, this would be problematic for lower frequencies. Please clarify.

Line487: Why is only a filter for γ specified? What is the exact meaning of n=3?

Line 489: The windows were wider for lower frequencies.

Line 490-491: Are these analyses actually reported?

Line 493 and following: Is the conditional GC analysis actually reported?

---

## [Author Response]

Essential revisions:1) GC Analysis using MUA (see below, rev1.1)

We performed the same GC analysis as in figure 7 using the MUA signal. The results are added as “Figure 7—figure supplement 4”. We added a section in the manuscript (line 264-266).

2) A more extensive of the distribution of receptive field sizes, and if appropriate, repeating the analysis when using 1 deg as reference (rev 1.2, editor comments).

We analyzed the RF diameter relative to their depth and plotted the results in “Figure 1—figure supplement 1”. We repeated the analysis using different stimulus sizes as reference (0.75 deg, 1.5 deg), and present the data in Figure 2 and Figure 4—figure supplements 1 and supplement 2. We edited a section in the manuscript where we link to these analyses (lines 93-94, lines 128-131, lines 188-189).

3) Clarify text regarding the difference between lateral intracolumnar feedback and top-down intra-area feedback (rev 2).

We have specified it more (line 27, 98-99, 287).

Reviewer #1:In this work, the authors study neural oscillations induced in macaque V1 by square-wave stimuli of varying sizes. The results suggest that α/β frequencys bands arise in L6, with long-range interactions to L2/3; and γ frequency arise in L4 or lower L3, with a series of short range interactions up the cortical column, all the way to L1. There is also an interaction from ~L4C to L6, for which the frequency changes depending on the size of the stimulus (low frequency for small and high frequency for large stimuli).I find the paper is worthy of publication in eLife as it stands, and could be made stronger with some additional work.1. The GC analysis was applied to four frequency-bands: theta, α, β, and low-γ. The same analysis could easily be applied to the MUA. Also, there is the possiblity of cross-frequency coupling (Spaak et al., 2012): high-frequency oscillations in one region may induce low-frequency oscillations in another (and vice-versa), and this could be studied with GC between different frequencies. Why is such analysis excluded from the paper? It would seem to be a natural fit, and I think this additional work would make the paper stronger.

We have now applied the analysis to MUA and added the results as “Figure 7—figure supplement 4” and added a brief text section to the manuscript (line 264-266) (see also Essential Revisions). Regarding a GC analysis between different frequencies, we have discussed this with Scott (rev#1) and external colleagues. Based on the discussion, the reviewer agreed that this is not necessary.

2. Much of the discussion throughout the paper focuses on the difference of effects with small centre-only stimuli and large stimuli which stimulate the surround. However, the distribution of RF size of the cortical columns being studied is not reported. This should be included so the size of the stimuli can be contextualised. I also posit that correcting for the size of the RF might lead to a clearer transition from the centre to surround recording condition. If there is a large range of RF sizes between sessions (a factor of 2 difference, or more) I ask that the authors consider this, or explain why correcting for the RF size would not be desirable.

We added an analysis of the RF diameters to the manuscript (Figure 1—figure supplement 1, see Essential Revisions). The median RF diameters across cortical depths ranged from ~0.7° to ~1.2°. An explicit correction for RF size would make sense if we would have aimed at describing the spatial integration properties of each cortical layer in isolation. However, we aimed at investigating the columnar oscillation pattern across layers. To nevertheless address the query in relative terms we performed key analyses with 2 additional reference sizes, namely 0.75 and 1.5 deg. The results are now shown in Figure 2 and Figure 4—figure supplements 1 and supplement 2. We added sections to the manuscript where we link to these analyses (lines 93-94, lines 128-131, lines 188-189). Based on these analyses we argue that the exact reference size is not key to the main results we try to convey.

3. On L160, the authors cite Einevoll to justify using the LFPbp, but I am sure he would instead advocate to use the CSD instead. Why use the LFPbp instead of the CSD?

We chose to use the bipolar LFP based on the results of Trongnetrpunya et al., (2016). They showed that connectivity analysis based on the bipolar referenced signal is less prone to noise and nearer to the ground truth of network connectivity than current source density signals. This is mentioned in the methods section in line 479-481.

We added the reference to Trongnetrpunya et al., (2016) Line 166 and Line 480

4. There are more accurate methods of estimating the CSD than the double-spatial derivative. There is the inverse-CSD (iCSD; Pettersen, 2006) and kernel-CSD (kCSD; Potworowski, 2012). Problems can arise with the double-spatial derivative when the current sources are small compared to the electrode measurement interval. These new methods take into account the electrode sampling interval, and the expected spatial extent of sources, in order to estimate the CSD from the LFP. Why use the double-spatial derivative instead of iCSD or kCSD?

We used the “standard” method from the iCSD-toolbox but did not mention it in the main text. We changed the text accordingly (line 463-473). We tested other methods from that same toolbox, but the results were not qualitatively different. The standard method was sufficient to extract the alignment criterion.

5. With regards to determining the spectral power (L483), why not use the Welch method, or similar, instead of just the FFT? This would give a more stable estimate of power, especially in the high frequencies.

We tested multiple spectral methods aiming at extracting the prominent ~40Hz oscillation evoked by spatial integration. We tried to avoid any methods that would result in a smoothing over spectral frequency, i.e. multitaper methods. On the other hand, in comparison with other methods, FFT seemed sufficient in quantifying the power of the 40Hz component.

6. It is curious that you find stimulus-driven changes centred at a frequency of 40Hz (Figure 4). Earlier works have found that stimulus-driven information is isolated into two compartments, one below 40Hz and the other above 40Hz (Belitski, 2008) and that frequencies below and above 40Hz encode information about different spatio-temporal components of stimuli (Lowe, 2017), whereas (in these works) frequencies around 40Hz do not contain information about the stimulus. Furthermore, they appear to be driven by dopaminergic neuromodulation (Zaldivar, 2018). Also, it is noteworthy that the stimulus presentation spatial frequency of 1.5cpd is the point where γ frequencies (<40Hz) become more informative about the stimulus than theta and α (Lowe, 2017).

Grating induced γ oscillations with a peak around 40Hz are a well described feature of V1 and have been frequently reported in the literature (Gray et al., 1989; Bauer et al., 1995; Gieselmann and Thiele, 2008; Jia et al., 2011). We agree that visual stimulation via natural images, as they have been used in Belitski et al., (2008), do not induce such a prominent 40Hz peak. However, we were particularly interested in the oscillatory interactions between cortical layers in the context of spatial integration which strongly modulates the 40Hz peak. A detailed analysis of frequency band specific information is beyond the scope of this study.

7. I am curious as to what other CSD-profiles were seen (L72). Were they just noisy, or indications that the probe was not aligned with cortical column? Or is there some (albeit weak) evidence CSD profiles are not uniform across V1? I assume the former, but ask in case there is a genuine minority CSD profile which multiple labs are seeing and rejecting. I have rejected a session before which was perfect, except for the inconvenient fact that the sign of all CSD currents were inverted.

This is an excellent question. But with the current dataset it is impossible to address it in a meaningful manner.

Reviewer #2:Anatomical tracing studies have established that the connectivity between visual areas abides by a global hierarchy, such that between any two visual areas, there is a graded hierarchical relation. Prior studies (Bastos, Neuron, 2015; Michalareas, Neuron, 2016) have shown that α and β GC is stronger in the top-down direction, whereas theta and γ GC is stronger in the bottom-up direction. Within a cortical column, the concept of a canonical microcircuit suggests a flow of information from granular to supragranular and then to infragranular layers. Prior studies (Livingstone, J. Neurophys., 1996; van Kerkoerle, PNAS, 2014) have shown that γ oscillations have characteristic laminar phase relations, with systematic lags as function of laminar distance from granular, and that α oscillations can show the opposite, i.e. systematic leads. These two sets of results, i.e. interareal and intracolumnar, are consistent, but need to be regarded separately. The hierarchical ordering of visual areas is supported by unequivocal evidence, whereas a columnar information-flow pattern is merely suggested, and different models are consistent with the empirical evidence. The current manuscript speaks to the columnar level and presents interesting evidence, partly consistent with van Kerkoerle, and partly extending the van Kerkoerle study to different stimulus sizes. It does not speak directly to the interareal level. The authors need to revise their manuscript to avoid confusion.Figure 6 and related figures and text: This is almost the same analysis as performed by van Kerkoerle, with the important difference that Kerkoerle used CSD, whereas the present manuscript uses bipolar LFP. The present manuscript makes direct comparisons to Kerkoerle and partly reports conflicting results. For a sound and fair comparison, the authors of the current manuscript need to provide these analysis based on the CSD.

We produced a new Figure 6—figure supplement 5 from our data to compare to figure 3 in Kerkoerle et al., (2014). And while these results are somewhat different to the LFPbp results, they still differ to those reported by van Kerkoerle et al., We believe that these differences arise from stimulus differences and possibly the use of a cognitively demanding task in van Kerkoerle's study. We now mention these differences in the text (lines 293-298, see also line 374-380).

The phase-lag and GC results are reported as mainly contradicting previous results. However, if only the results are considered that use exactly the same methods as previous studies (e.g. Kerkoerle with the same stimulus size), results are not so differet. Results obtained with other stimulus sizes are not in conflict with this. And all results of those intra-V1 recordings are not in conflict with interareal GC analyses. Thus, the authors should limit reported conflicts to aspects that can really be compared.

We now mention possible sources for these differences and acknowledge that the use of large grating stimuli could be a key element (line 293-298). However, van Kerkoerle used large stimuli (full field structured background, and elongated lines (curve tracing task)) so comparing that directly to small grating stimuli is equally not really compatible.

The same holds for the reporting of power. For example line 307 – 309: Xing, 2012 did show γ power in many layers, with a distinct peak in deep L5/upper L6; and Buffalo, 2011 focused on spike-field coherence. SFC is a very different metric from LFP power. It is possible that a particular layer, for a particular stimulus size, shows clear γ LFP power, but maybe the spikes there do not phase-lock to the LFP γ phase, such that γ SFC is low. Thus, again, the authors should limit reported conflicts to aspects that can really be compared.

We have changed the wording and the associated referencing, to account for those differences (line 332-333).

I am somewhat concerned about the upwards dominance of GC (Figure 7 supp 1). Could there be any trivial reason for this?

We do not think this to be the case. We have reported a similar pattern in a recent publication (Ferro et al., 2021, PNAS), where animals engaged in an attention demanding task, and where we simultaneously recorded in V1 and V4. Here V1 showed the pattern, while V4 did not. We now cite that study (line 248). It has been reported that granger causal interactions depend on absolute power of the signal, where the more powerful signal could be seen as the driver. However, this is not the case for our data, where absolute power dominates in supragranular layers.

Line 326: Early work from the Eckhorn lab showed clear γ in V2. A recent preprint from the Richard Born lab shows strong reductions in V1 γ upon V2 cooling. Also, Bastos, Neuron, 2015 showed clear V2-to-V1 γ GC; note that this paper relates the DAI to the SLN in a graded manner, and the known feedforward-type (supragranular originating) projections from V2 to V1 are expected to carry γ in this top-down direction. Thus, we do know that the input feedback signal shows γ.

We now reference these studies and list them as a potential main source of the layer 5 γ frequency sources with larger stimuli (lines 350-354).

Figure 4D: It would be very helpful to see spectra (cross-cuts through the presented spectro-depth plots), at the depths with strongest γ, i.e. around 450 and -450

We added Figure 4—figure supplement 3 and referred to it in the respective Results section (line 164-189).

Figure 7 and similar plots: This is a matter of taste, but I find a matrix version of such results much easier to parse.

Newly added Figure 7—figure supplement 4 shows a matrix view of the data.

Line 37,38: Interareal theta-band GC is stronger in the feedforward, not the feedback, direction (Bastos, Neuron, 2015; Spyropoulos, PNAS, 2018). The feedforward nature of theta is actually correctly referred to in line 144.

We changed this (line 41)

Line 44: Classical work by Lopes da Silva located the α source rather deep (Lopes da Silva, Neuroscience Letters, 1977, pages 237 – 241).

We now added this reference (line 39)

Lines 69-74: Clarify whether all three conditions have to be met, or whether one suffices.

We edited that section in the text (line 68-74) in reply to a point raised by reviewer#1.

Line 104: How was the ANOVA done? Is it suitable to provide the p-value for the statement in the preceding sentence on 3 ms longer latencies?

We performed a one-factor-ANOVA across stimulus sizes to estimate if there was a main effect of stimulus size. We rephrased the section describing the results to avoid the impression that we statistically tested the time differences indicated by the plot (line 105-108).

Line 107: The figure lists p=0.98, whereas the text lists p=0.59. Also, the text specifies a one-way ANOVA. Please clarify.

The p-value refers to the one-factor-ANOVA across all stimulus sizes (see previous comment). We changed the p-value to the correct value of p=0.98 as shown in the figure (line 111).

Line 169: Gieselmann and Thiele previously showed an actual γ peak for small stimuli, whose peak frequency decreased with stimulus size. This should be related to the current results.

In response to the comments for Figure 4D of reviewer#2 we plotted the spectral power at selected depth positions (new Figure 4—figure supplement 3). These plots qualitatively show an inverse relationship between the γ peak frequency and stimulus size similar to the dataset described in Gieselmann and Thiele (2008). See references to the new figure in (line 164-189).

Figure 4B, D and Figure 5A, D, G, J: The text and figure legends are quite clear on that these plots show power changes due to stimulation. I am surprised that changes are exclusively positive. Previous studies have found stimulus induced power decreases, e.g. in α. Are these rectified power changes? In any case, the authors should address this excplicitly, simply to avoid confusion. Also, where the text refers to those panels, it should be careful to actually refer to power changes (not just power).Line 177 and a few other places: the Xing, Shapley, PNAS, 2012 paper showed several γ peaks along cortical depth, including a weak one at lower layer 5/upper layer 6.

In our passively fixating monkeys, we indeed rarely found stimulus induced α decreases, that is below levels of spontaneous activity (negative z-score). However, large stimuli induced less power for low frequencies while higher frequencies were simultaneously induced with more power (Figure 4, 5 and Figure 4-supplement 3).

We introduced the term “induced spectral power” (iSP) in line 171 and used it in the section describing the results if figure 4 and 5 (line 164-189).

We agree that Xing et al., (2012) have also shown increased γ power in infragranular layers and added a reference in line 185.

Line 193: I was surprised by the use of the term phase-amplitude-coupling for this analysis. Usually, PAC is used for the relation of low-frequency phase to higher-frequency power, not the amplitude of the raw signal.

We agree that we misused the term phase-amplitude-coupling and changed the terminology throughout the paper, using the terms “phase-triggered-average” (PTA) or “phase-coupled” were appropriate.

Line 220-221: This statement is true for one isolated phase lag value. However, a pattern of phase lags as analyzed by Kerkoerle or in the present manuscript (or in the context of phase-lag spectra) resolves these uncertainties. However, the statement can be made that phase relations are sometimes not consistent with GC (explicitly mentioned by Brovelli, PNAS, 2002, start of page 9853), and this can be resolved with GC methods (Witham, Riddle, Baker, J Physiol, 2011).

We have changed the text accordingly (lines 232-235)

Line 264: Figure 7 supp 1 B shows the dominant influence of infra onto supra to be around 20 Hz, i.e. in the same β band that Bastos, Neuron, 2015 found to be feedback related. Figure 7 supp 1 G shows infra to very supra and shows a broad α peak that is at least compatible with Buffalo, PNAS, 2011.

We have changed our discussion in line with these comments (lines 283-288).

Line 301, 302: Also Peter, Fries, Vinck, eLife, 2019.

We have added the reference (line 328).

Line 335: Barone, JNS, 2000 shows an SLN of 47 %, i.e. not predominantly supragranular, and most likely not predominantly from just layer 3.

We now acknowledge this and cite Barone et al., (line 361-362).

Line 339-342: Top-down projections, even if short-range, always have a component of feedback-type connectivity, targeting deep layers. Thus, no reason to invoke longer-range feedback connections.

While we do not dispute the short-range feedback terminating in deeper layers (and we now make this explicit, line 361-362, the feedback from V2 should predominantly affect supragranular layers), which is not what we see. Also, the size effects seen in our data suggest that areas with larger RFs are involved.

Line 483: How do the authors obtain 1000 samples in 100 ms? Are they transforming a 10 kHz signal? If all FFT analys were based on 100 ms windows, this would be problematic for lower frequencies. Please clarify.

Data from 100 ms sliding window were padded to 1024 points by MATLAB’s “spectrogram” function to achieve a high spectral resolution. We agree that the window width may cause absolute power estimates in lower frequencies (< 10 Hz) to be inaccurate. However, we would argue that the conclusions drawn from Figure 5 C, F still hold as we were comparing induced power levels between different stimulus sizes, and our focus is generally on frequencies >10Hz.

Line487: Why is only a filter for γ specified? What is the exact meaning of n=3?

We used the same procedure to extract the troughs of oscillations in the four frequency bands shown in figure 6 (4-8,8-13,14-25,30-55 Hz). N is the order of the butterworth filter we used as a bandpass filter. We edited the respective method section accordingly (line 515-521).

Line 489: The windows were wider for lower frequencies.

Yes, they were. We edited the respective method section accordingly (line 515-521).

Line 490-491: Are these analyses actually reported?

No, we didn’t report quantification of PTA coupling and removed the respective text in the methods (line 515-521).

Line 493 and following: Is the conditional GC analysis actually reported?

No, it is not reported. We removed the respective text from the method section (line 522-528).